# Sub-continental-scale carbon stocks of individual trees in African drylands

Compton Tucker[1✉], Martin Brandt[2,3✉], Pierre Hiernaux[2,4✉], Ankit Kariryaa[2,3,5], Kjeld Rasmussen[3], Jennifer Small[1,2], Christian Igel[5], Florian Reiner[3], Katherine Melocik[1,2], Jesse Meyer[1,2], Scott Sinno[1,2], Eric Romero[1,2], Erin Glennie[1,2], Yasmin Fitts[1,2], August Morin[1,2], Jorge Pinzon[1,2], Devin McClain[1,2], Paul Morin[6], Claire Porter[6], Shane Loeffler[6], Laurent Kergoat[7], Bil-Assanou Issoufou[8], Patrice Savadogo[9], Jean-Pierre Wigneron[10], Benjamin Poulter[1], Philippe Ciais[11], Robert Kaufmann[12], Ranga Myneni[12], Sassan Saatchi[13] & Rasmus Fensholt[3]

The distribution of dryland trees and their density, cover, size, mass and carbon content are not well known at sub-continental to continental scales[1–14]. This information is important for ecological protection, carbon accounting, climate mitigation and restoration efforts of dryland ecosystems[15–18]. We assessed more than 9.9 billion trees derived from more than 300,000 satellite images, covering semi-arid sub-Saharan Africa north of the Equator. We attributed wood, foliage and root carbon to every tree in the 0–1,000 mm year$^{-1}$ rainfall zone by coupling field data[19], machine learning[20–22], satellite data and high-performance computing. Average carbon stocks of individual trees ranged from 0.54 Mg C ha$^{-1}$ and 63 kg C tree$^{-1}$ in the arid zone to 3.7 Mg C ha$^{-1}$ and 98 kg tree$^{-1}$ in the sub-humid zone. Overall, we estimated the total carbon for our study area to be 0.84 (±19.8%) Pg C. Comparisons with 14 previous TRENDY numerical simulation studies[23] for our area found that the density and carbon stocks of scattered trees have been underestimated by three models and overestimated by 11 models, respectively. This benchmarking can help understand the carbon cycle and address concerns about land degradation[24–29]. We make available a linked database of wood mass, foliage mass, root mass and carbon stock of each tree for scientists, policymakers, dryland-restoration practitioners and farmers, who can use it to estimate farmland tree carbon stocks from tablets or laptops.

Improved knowledge of dryland trees, defined here as having a green crown area >3 m$^2$ with an associated shadow (Extended Data Fig. 1), is essential to understand their roles in local livelihoods, economies, ecosystems, the global carbon cycle and the climate system in general. Basic information about the distribution of dryland trees and their density, cover, size, mass and carbon content are not well known[2–5]. This knowledge is required for understanding the functional traits of trees in relation to water resources with changes in climate, predicted increase in aridity and the number and duration of drought events[30,31]. The sources of information used to estimate carbon stocks in drylands include field surveys at plot scale; ecosystem models[23]; and low-resolution, moderate-resolution and high-resolution satellite images[4–14], which are used to infer bulk properties such as averages of tree cover, dry masses and carbon density per unit area at a much coarser spatial scale than individual trees.

Although most emphasis is put on the development of advanced monitoring techniques for forested ecosystems, none of these sources combine wide/total coverage and representation of each individual tree[5]. Reaching this level of detail is critical for dryland monitoring and management because dryland trees grow isolated and in highly variable size and density. Most current studies producing or using areal averages of tree cover, wood mass or carbon stocks in drylands are either at the very local level[12] or the information for drylands is derived from global maps[13], which are rarely trained and validated in drylands and often apply the same method on both forests and dryland vegetation[6–8]. Although national tree inventories exist for few dryland countries, the amount of labour required and their uncertainty are high. As a result, all existing assessments on dryland carbon stocks are highly uncertain, very difficult to validate and do not provide the means for a detailed characterization at the level of individual trees[14]. Furthermore,

[1]Earth Science Division, NASA Goddard Space Flight Center, Greenbelt, MD, USA. [2]Science Systems and Applications, Inc., NASA Goddard Space Flight Center, Greenbelt, MD, USA. [3]Department of Geosciences and Natural Resource Management, University of Copenhagen, Copenhagen, Denmark. [4]Pastoralisme Conseil, Caylus, France. [5]Department of Computer Science, University of Copenhagen, Copenhagen, Denmark. [6]Learning and Environmental Sciences, University of Minnesota, Saint Paul, MN, USA. [7]Géosciences Environnement Toulouse, Observatoire Midi-Pyrénées, UMR 5563 (CNRS/UPS/IRD/CNES), Toulouse, France. [8]Dan Dicko Dankoulodo University of Maradi, Maradi, Niger. [9]FAO Subregional Office for West Africa, Dakar, Senegal. [10]ISPA, UMR 1391, INRAE Nouvelle-Aquitaine Bordeaux, Villenave d'Ornon, France. [11]Laboratoire des Sciences du Climat et de l'Environnement, CEA-CNRS-UVSQ, CE Orme des Merisiers, Gif sur Yvette, France. [12]Department of Earth & Environment, Boston University, Boston, MA, USA. [13]Jet Propulsion Laboratory, California Institute of Technology, Pasadena, CA, USA. ✉e-mail: compton.j.tucker@nasa.gov; mabr@ign.ku.dk; pierre.hiernaux2@orange.fr

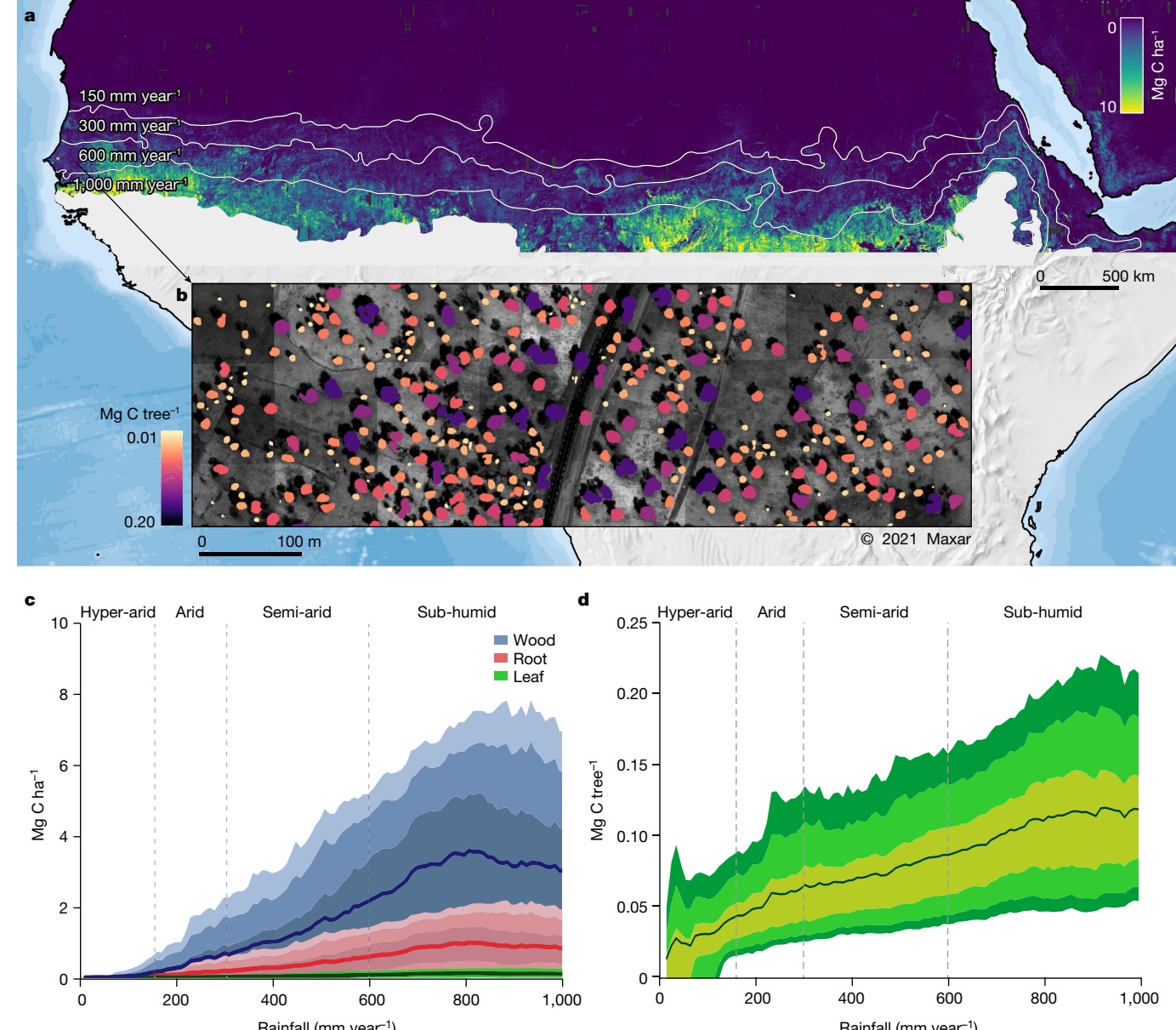

**Fig. 1 | Wood, foliage and root carbon of 9,947,310,221 trees with crown area >3 m² across 9.7 million km² were mapped. a,** Our study covered the southern Sahara, the Sahel and the northern Sudanian zone of Africa and showed the aggregated carbon density (foliage + wood + root) per hectare for 9,947,310,221 tree crowns from the 0–1,000 mm year⁻¹ mean precipitation area. The isohyets mark the 150, 300, 600 and 1,000 mm year⁻¹ rainfall zones

(from north to south). **b,** Example showing the woody carbon stock of each single tree for an agroforestry area in Senegal. **c,** Mean tree carbon density at the 5th, 25th, 75th and 95th percentiles along the rainfall gradient for wood, foliage and root carbon. **d,** Mean carbon stock of individual trees at the 5th, 10th, 25th, 75th, 90th and 95th percentiles along the rainfall gradient. Our definition of a tree is a green leaf crown >3 m² with an associated shadow (Extended Data Fig. 1).

the contribution of different dry-mass components—wood, foliage and root mass—to the overall carbon stock is unknown at large scales.

At the same time, it remains unknown whether ecosystem models quantify the right amount of carbon and the lack of validation of global models or maps in dry areas fuels narratives of possible underestimation or overestimation of carbon stocks of drylands and their role in accelerating or mitigating climate change[12,18]. The missing information on trees at the level of individuals is decisive for improved management of woody resources in drylands: to accurately monitor deforestation spurred by clearing of trees for cropping, mining, infrastructure and urban development[24]. Furthermore, accurate monitoring of the tree resource at the level of individual trees is instrumental for tree-planting initiatives, for reporting the correct number of trees and carbon stocks for national reporting schemes, such as the Paris Agreement, or to have

a reliable system that allows payments for environmental services to farmers and villages. Although deforestation and afforestation areas can be accurately mapped using current methods and data in forest ecosystems, no monitoring system exists for trees outside forests and their carbon pools[32].

At present, large amounts of funding are being allocated to dryland-restoration activities and the monitoring of success or failure is based on local inventories lacking large-scale assessments of survival rates of planted trees. The Great Green Wall of the Sahara and the Sahel initiative has recently been subject to renewed interest and increased investments[33–35]. This initiative was conceived to address the increasing challenges of desertification and drought, food insecurity and poverty in the wake of climate change. Yet the tracking of projects and their successfulness remains a great challenge, as no

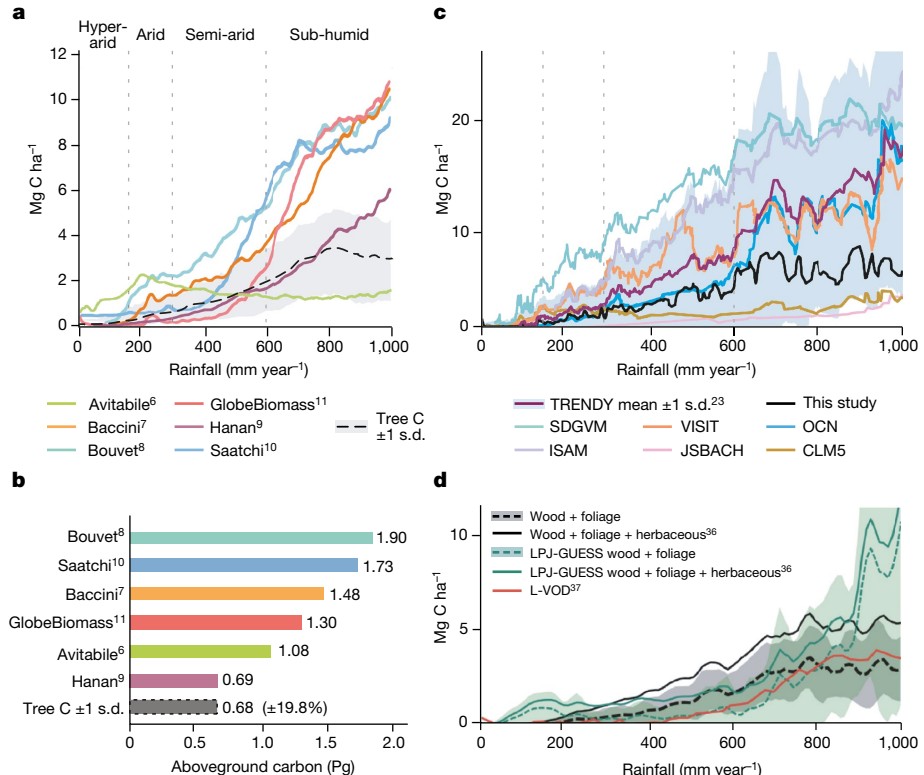

**Fig. 2 | Comparisons between current aboveground carbon-density maps and our estimations derived from 9.9 billion trees. a**, Aboveground carbon density from state-of-the-art maps using satellite data[6–11]. Tree carbon from this study is derived from wood + foliage carbon plotted with ±1 standard deviation in the grey zone. **b**, Aboveground carbon stocks aggregated over the 0–1,000 mm year⁻¹ rainfall zone. Our estimations (grey colour) of 0.68 Pg are wood + foliage carbon. The combined uncertainty from neural net area mapping, tree crown omission and commission errors, and allometric conversion of tree crowns into tree wood, foliage and root carbon was ±19.8% (Methods). **c**, Vegetation carbon density from the mean of 14 TRENDY dynamic ecosystem

models and data from six individual TRENDY models for aboveground and belowground carbon[23] are compared with our tree carbon with aboveground herbaceous carbon added from passive microwaves[36]. **d**, Aboveground carbon density from the LPJ-GUESS model[23], selected here as it uses trees outside the prescribed forest fraction, and our estimations are compared along the rainfall gradient. L-VOD[37] was converted to carbon density using coefficients from a linear correlation with our map (Extended Data Fig. 4). Aboveground herbaceous carbon was derived from ref. [36]. The sample size for the 0–1,000 mm year⁻¹ rainfall zone was 9,947,310,221 tree crowns >3 m².

monitoring system is in place. Equally important, large-scale monitoring of single trees will create a foundation for establishing improved knowledge on the functional traits of dryland trees, such as survival, growth and mortality, controlled by the complex interplay between biotic and abiotic factors[32]. Afforestation initiatives should also be rooted in a solid ecological understanding of the local environment to avoid causing water shortages for small-holder farming systems[33].

The combined use of very-high-resolution satellite images and artificial intelligence made it possible to identify isolated trees and map their crown area at large scales, covering the western Sahara–Sahel–Sudan areas[1]. This approach of mapping individual trees has been extended to a 7.5-times-larger area covering the drylands across Africa, from the Atlantic Ocean to the Red Sea from 9.5° N to 24° N latitude between the 0 and 1,000 mm year⁻¹ isohyets, using 326,523 satellite images at a 50-cm spatial resolution, and coupled with machine learning to map 9.9 billion trees (Fig. 1 and Methods). The large-scale mapping of individual tree crowns provides an unprecedented opportunity to apply allometric equations to estimate carbon stocks derived from foliage, wood and root dry masses at local scales to large regions, here close to 10,000,000 km² (Extended Data Fig. 2). We take this step to assess the woody carbon pool by adding up tree-by-tree values, calculated using allometric equations to predict foliage, wood and root dry masses from crown area multiplied by the average carbon concentration (0.47). These allometric equations were established by destructive sampling of trees from 26, 27 and 5 species, respectively, selected within a rainfall gradient from 150 to 800 mm year⁻¹. Comparisons with allometric

equations established in wetter tropical areas ensure applicability of these equations to wetter zones, at least up to 1,000 mm year⁻¹ rainfall[19]. We estimated the combined uncertainty from the allometric equations and the tree crown detection to be ±19.8%.

The information of carbon stocks of 9.9 billion trees is compared with a set of state-of-the-art TRENDY ecosystem models[23] as well as current satellite-observation-based regional carbon stock maps[6–11]. We introduce a publicly available 'viewer', which allows farmers, villagers, policymakers and all stakeholders to retrieve the foliage wood and root masses and the corresponding carbon stock of each tree using a mobile device. We expect that this could improve not only the amount of information available but also the reporting and monitoring of trees and their carbon stocks at various scales, from the individual field plot to country scales.

## Carbon stocks at the tree level

We applied a deep-learning-based tree mapping on a large number of satellite images and measured 9,947,310,221 tree crowns: all woody plants with a shadow and a crown area >3 m² from the hyper-arid (0–150 mm year⁻¹), arid (150–300 mm year⁻¹), semi-arid (300–600 mm year⁻¹) and dry sub-humid (600–1,000 mm year⁻¹) rainfall zones of tropical Africa north of the Equator and south of the Sahara (Fig. 1). The average carbon stock of a single tree is 51 kg C in the hyper-arid, 63 kg C in the arid, 72 kg C in the semi-arid and 98 kg C in the sub-humid zone. The individual tree information was projected

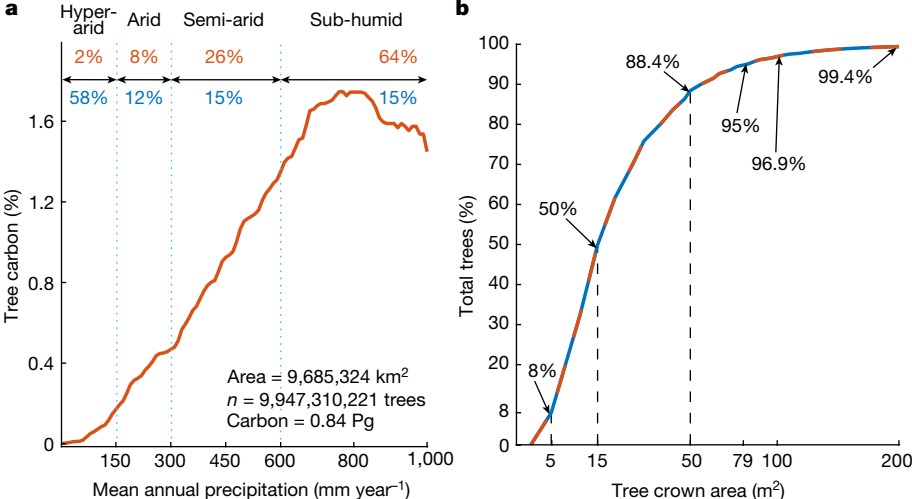

**Fig. 3 | Precipitation, tree carbon and crown area. a**, The tree carbon probability density function computed along the rainfall gradient of the study from the hyper-arid (0–150 mm year⁻¹), arid (150–300 mm year⁻¹), semi-arid (300–600 mm year⁻¹) and dry sub-humid (600–1,000 mm year⁻¹) rainfall zones. The percentage area of each semi-arid zone is shown in blue and the percentage of total carbon in red. The increasing tree carbon probability function shows the importance of precipitation for tree carbon in semi-arid regions. Most tree carbon is found in the semi-arid (26%) and dry sub-humid zones (64%), which represent only 30% of the area in our study. The per cent carbon density contribution by rainfall zones is linearly related to the tree carbon density (Mg C ha⁻¹) reported in Fig. 1c by a factor of 2.5. **b**, A total of 88.4% of our mapped trees had crown areas <50 m². The average tree crown area in the 0–150 mm year⁻¹ zone was 15.1 m², for the 150–300 mm year⁻¹ zone it was 18.4 m², for the 300–600 mm year⁻¹ zone it was 20.9 m² and for the 600–1,000 mm year⁻¹ zone it was 28.1 m². Only 11.6% of our mapped trees had crown areas >50 m² and less than 0.6% had crown areas >200 m².

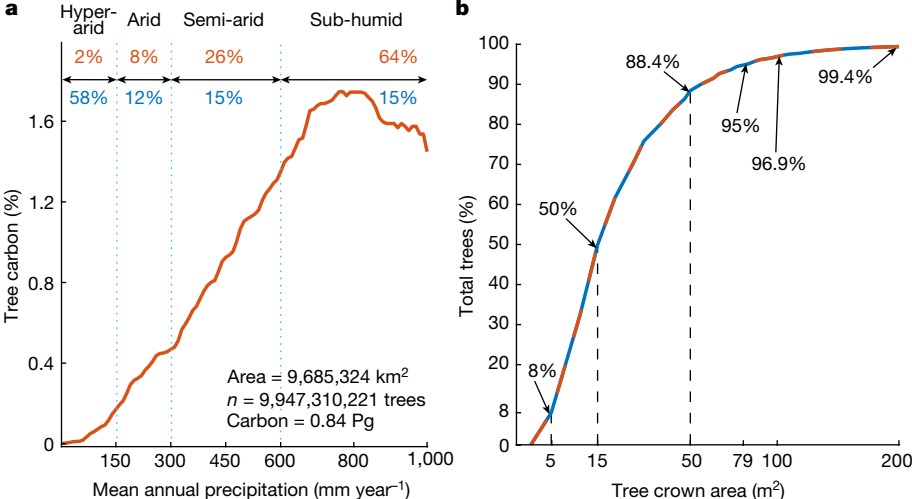

**Fig. 4 | Different components of the viewer.** This example shows Widou Thiengoly in semi-arid Senegal surrounded by tree plantations, which are partly related to a Great Green Wall[34] project aiming to increase tree density and improve livelihoods in the Sahel. **a**, Tree crown segmentations from the neural net mapping. **b**, Wood, foliage and root carbon calculated for each tree (Methods). **c**, Carbon density per hectare aggregated from carbon stocks of single trees to the hectare scale. **d**, Our viewer includes all information from **a** to **c**. This online tool provides information on crown area; foliage, wood and root carbon of single trees; and aggregates carbon to the hectare scale. These data can be accessed by policymakers and stakeholders to monitor areas of interest. The viewer can be accessed at https://trees.pgc.umn.edu/app.

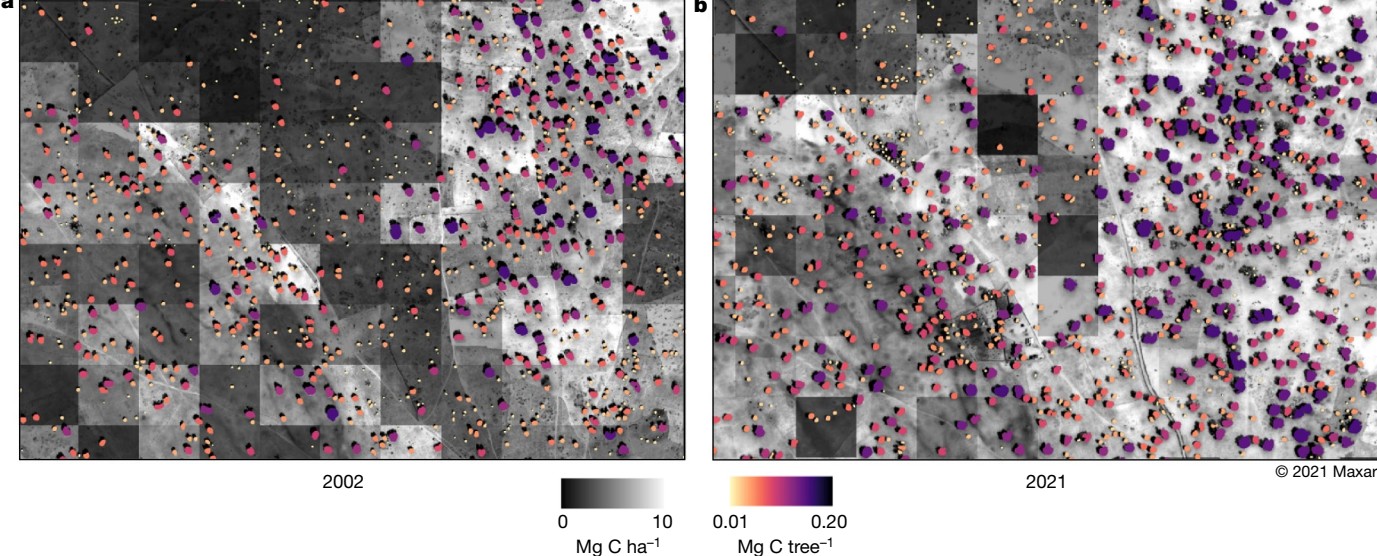

0 ____ 10
Mg C ha⁻¹

0.01 ____ 0.20
Mg C tree⁻¹

**Fig. 5 | Monitoring at the level of single trees from Khombole, Senegal.**
**a**,**b**, A 50-cm-scale image from 2002 (**a**) and a 50-cm-scale satellite image from 2021 (**b**) showing an agroforestry area at the same location. Tree cover has increased between 2002 and 2021 and the average carbon density of both areas was calculated and increased from 6 to 10 Mg ha⁻¹. A large number of trees grow on farmlands, keeping the soils fertile and reducing the need for fallow periods. The greyscale of the background images indicates the carbon density per hectare, whereas the colour scale shows the carbon content of individual trees. This is a good example of the tree restoration monitoring potential in our study area.

to the area by calculating the carbon density in Mg C ha⁻¹, which was on average 0.03 Mg C ha⁻¹ in the hyper-arid, 0.54 Mg C ha⁻¹ in the arid, 1.54 Mg C ha⁻¹ in the semi-arid and 3.73 Mg C ha⁻¹ in the sub-humid zone. Although foliage mass has a small overall fraction of the total dry mass (3%), it is an important variable for quantification of browse potential and serves as a proxy for other ecosystem processes, such as transpiration, photosynthesis and nutrient cycling. The proportion of root mass is on average 15–20% of the total mass.

## Current carbon map and model comparisons

We compared our aboveground carbon-density maps (foliage + wood) derived from individual trees with current state-of-the-art maps (Fig. 2 and Extended Data Fig. 3) available at moderate spatial resolutions of 30–1,000 m. The temporal dynamics were assessed by low-frequency passive microwaves (L-VOD)[36,37], which has emerged as a tool for the assessment of carbon stock dynamics at the 25 × 25-km spatial scale (Extended Data Fig. 4). Moreover, we compared carbon-density maps and dynamics with dynamic ecosystem models from the TRENDY database with a 50 × 50-km grid cell size[23]. None of these maps were designed specifically for drylands; most dynamic ecosystem models and satellite-based models are developed and trained for forest ecosystems and, in the case of the TRENDY models, used meteorological forcings and prescribed vegetation maps that contain further uncertainties for comparative purposes.

Existing carbon-density maps compare differently with our assessment based on individual trees and there is little spatial agreement among the maps (Fig. 2a,b). Notably, although areas of scattered trees having a relatively low carbon density are largely mapped as zero carbon in previous maps except for ref. [9], areas of denser tree cover and some areas typically without trees, such as wetlands, irrigated croplands and desert mountains, have considerably higher values than our assessment. This leads to an overall higher carbon stock of the area compared with our results. Although we do not map herbaceous vegetation in our study, the tree cover we map can be used to disaggregate herbaceous vegetation from trees (Extended Data Fig. 5).

At regional scales, dynamic ecosystem model vegetation carbon shows a considerable variability, but the mean follows our estimates of herbaceous, wood, foliage and root carbon along the rainfall gradient (Fig. 2c). Notably, whereas previous studies assumed that ecosystem models underestimated dryland carbon stocks, our results show overall higher values from the model outputs as compared with the assessment based on individual trees, although large variations between models exist. Only considering aboveground carbon, the example of LPJ-GUESS shows slightly lower values than our assessment up to about 800 mm year⁻¹ rainfall (Fig. 2d).

Both ecosystem models and previous satellite-based carbon maps diverge markedly from our results beyond 700–800 mm year⁻¹ rainfall. All other maps assume a continuous increase beyond this rainfall zone, yet our results reach a plateau at 800 mm year⁻¹ and no further increase in carbon is observed with higher rainfall up to 1,000 mm year⁻¹. We acknowledge that the uncertainty of our results increases with denser canopy cover and that we miss all understory vegetation. However, statistical evaluations of the rainfall–tree density relationship from our data indicate that neither carbon stocks per tree (Fig. 1d) nor tree cover further increased between 800 and 1,000 mm year⁻¹ rainfall (Fig. 3a). Trees with crown area <50 m² make up 88% of the total number of trees, whereas trees in the semi-arid and sub-humid zones constitute 90% of the total carbon in our study (Fig. 3).

## Application at the tree level

The comparison with dynamic global vegetation models and existing biomass maps shows some similar patterns at coarse scale, yet none of these maps can be used to derive information at the level of individual trees needed to support policymakers and decision-makers. For this reason, we introduce a viewer (Fig. 4), which is built on Mapbox and OpenStreetMap, and can be accessed online by everyone and from anywhere. The viewer includes all 9.9 billion trees as objects, and the wood, foliage and root mass can be accessed individually for each of them. As an example, we show the area of Widou Thiengoly, an area in Senegal in which tree planting for the Great Green Wall has been promoted over the past decades (Fig. 4a). Although previous assessments on the success of tree plantations were based on narratives, visual interpretations or site visits, the viewer provides an unbiased tool for evaluating success and failure of initiatives, as well as quantifying the

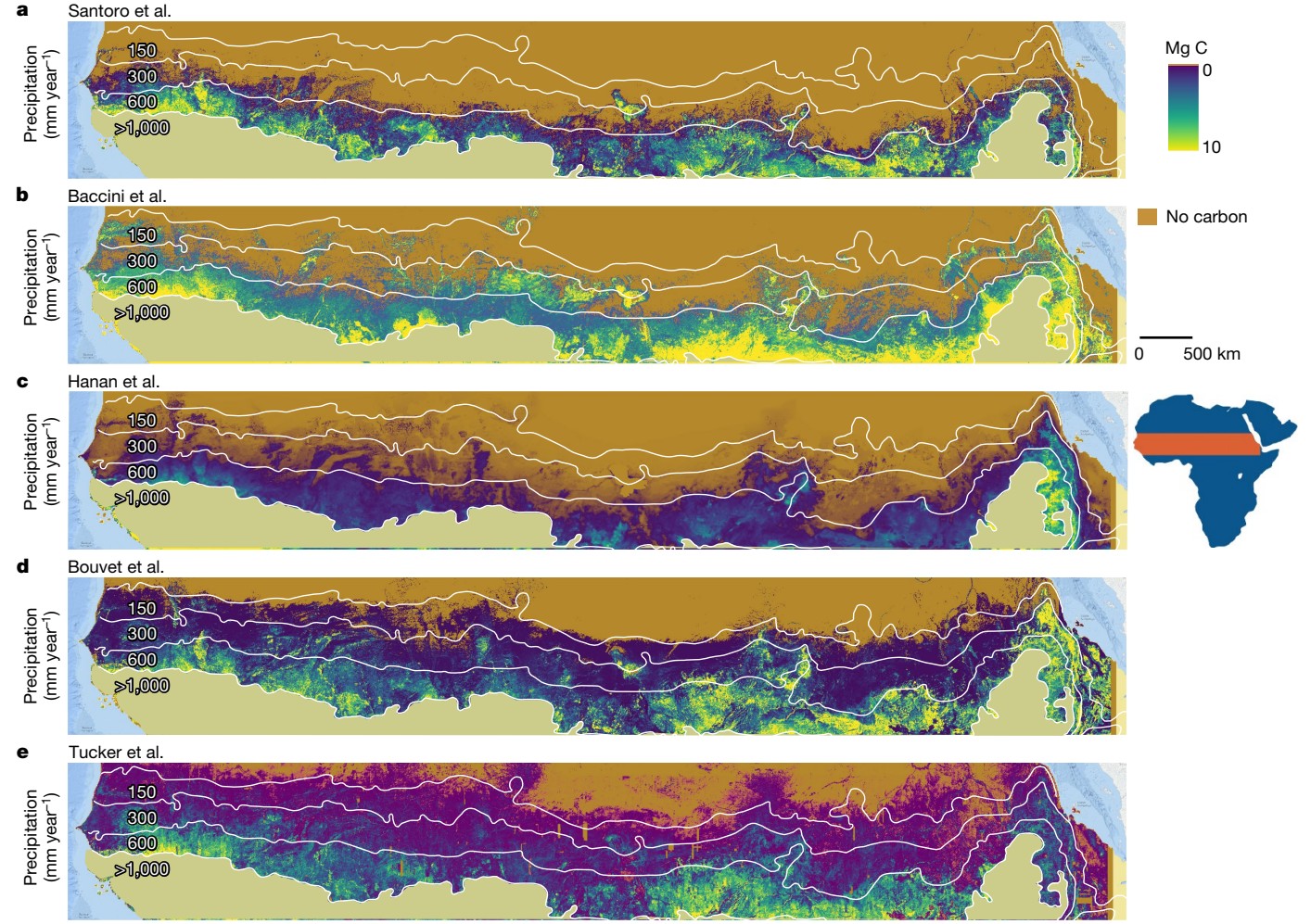

**Fig. 6 | Comparison of different aboveground carbon-density maps.**
We show the aboveground carbon density for our study area compared with different sources. Areas beyond 1,000 mm year[-1] rainfall are masked out. Data are from Santoro et al.[11] (**a**), Baccini et al.[7] (**b**), Hanan et al.[9] (**c**), Bouvet et al.[8] (**d**) and Tucker et al. (this paper; foliage + wood) (**e**). See also Fig. 2a and Extended Data Fig. 3.

carbon stocks gained by each planted tree or lost by each removed/dead tree. The example shown in Fig. 4 illustrates that high-density plantations in this arid region reach carbon-density values of about 5 Mg C ha[-1] (Fig. 1c), but the survival rate of planted trees has been a long-lasting concern that needs to be carefully monitored to be able to assess the efficacy of Sahelian tree-planting programmes.

Another example shows an agroforestry region in Senegal, north of Khombole that has a relatively high density of trees, which has increased the carbon stocks of the region considerably. The example area shown in Fig. 5 has almost doubled carbon density between 2002 and 2021 (Fig. 5).

## Discussion

Our assessment is a large-scale estimation of wood, foliage and root carbon at the level of individual trees. The finding that global ecosystem models and previous carbon-density maps estimate higher carbon stocks in African drylands compared with our assessment based on 9.9 billion individual trees seems surprising, as current tree-cover maps are not able to correctly account for scattered trees and thus should considerably underestimate the number of trees in these areas[1]. The explanation for this apparent paradox—higher tree cover but less carbon—is related to the fact that previous models are rarely developed, trained and validated with plots of very sparse tree cover, thus leaving high

uncertainty for drylands with scattered trees. Consequently, areas with scattered trees are often represented by zero values (Fig. 6), whereas the carbon density of larger groups of trees may be overestimated in previous assessments, as these areas are wrongly considered as dense forests. In essence, most previous assessments do not accurately map carbon density below 10 Mg C ha[-1], if at all, and may overestimate the carbon stocks of dryland 'forests'. Moreover, if the region is taken as a whole, green crops and herbaceous vegetation affect optical images, whereas steep topography and wetlands/irrigated areas affect the radar backscatter, both predicting higher carbon stocks than our estimations. Although we used allometric equations specifically developed from locally sampled field data[19], 95% of the trees we mapped had a crown area <78 m[2]. This introduces a small uncertainty in carbon values for the 5% of tree crowns >78 m[2] in more humid areas, where trees are taller and/or larger.

Nevertheless, the divergence between our results and previous assessments in higher-rainfall zones needs to be further investigated and our maps should be used with caution beyond 800 mm year[-1] rainfall. The indirect inclusion of the tree height and the application of the same equation to all tree species are uncertainty factors that will be assessed in future versions of the dataset. Finally, the fact that larger trees shade out smaller trees in areas of dense tree cover makes the method based on individual tree counting less suited to more humid areas.

Herbaceous dry mass can contribute considerably to the annual carbon density. However, most herbaceous plants of the region are annuals that die off each year and do not constitute a residual carbon stock but have a high inter-annual variability. The herbaceous mass used in our study[36] shows the seasonal peak value, which drops by about 25% within only a few weeks (Extended Data Figs. 1a and 5). Traditionally, remotely sensed separation of herbaceous vegetation from woody foliage is challenging with both optical and radar satellite data. We overcome this by measuring individual tree crown areas.

The carbon difference between ecosystem models and our study can be explained by different forest fractions assumed by each model (Extended Data Fig. 6). Most of the dynamic global vegetation models do not simulate trees outside forests and woody carbon is usually a sum of predefined forest areas. Differences may also result from a simplistic implementation of disturbances, in particular, fire, grazing and the fact that we did not include belowground herbaceous carbon in our estimates. Still, the results of the dynamic vegetation models are closer to our estimations than previously assumed and the inclusion of our data may improve future modelling results, leading to more realistic forecasts of the impact of climate change on drylands.

Dryland trees are not only a carbon stock but also provide ecosystem services that are valuable to the environment and support local livelihoods, including timber, fuel wood, protection against soil erosion and loss, soil fertilization, shade and nutrition for tree crops[15]. The benefits of increased tree cover are many and establishing an operational monitoring system for dryland trees is critically needed. The dynamics of growth and mortality of trees outside forests goes undetected by conventional monitoring systems based on satellite imagery with a spatial resolution >10 m. Although our current assessment at the level of individual trees does not yet include a temporal dimension (except for the exemplary case provided in Fig. 5), it is a baseline of the number, mass and carbon stock of trees outside forests at the sub-continental scale. The publicly available viewer makes this information accessible for scientists, policymakers, stakeholders and individual farmers, who can easily quantify woody carbon stocks of a given area, down to the level of a single tree growing in a private yard.

A next step will be adding a temporal dimension to the wall-to-wall mapping we describe and we expect it to be possible from this source of data, at least with decadal time steps. This will facilitate addressing the impact of droughts, restoration and policies at various scales, down to the level of individual trees. High spatial resolution is the key to improved tree inventories in drylands. The ever-increasing availability of satellite images will make continental-scale assessments of carbon pools and dynamics at the individual tree level realistic in near-real time. This will be key to developing robust schemes for dryland management plans needed to achieve the United Nations' Sustainable Development Goals. Our paper is a step in that process.

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

## Methods

### Overview

This study establishes a framework for mapping carbon stocks at the level of individual trees at a sub-continental scale in semi-arid sub-Saharan Africa north of the Equator. We used satellite imagery from the early dry season (Extended Data Fig. 1). The deep learning method developed by a previous study[1] allowed us to map billions of discrete tree crowns at the 50-cm scale from West Africa to the Red Sea. Then we used allometry to convert tree crown area into tree wood, foliage and root carbon for the 0–1,000 mm year$^{-1}$ precipitation zone in which our allometry was collected (Extended Data Fig. 2). We introduce a viewer that enables the billions of trees to be viewed at different scales, with information on location, metadata of the Maxar satellite image used, tree crown area and the estimated wood, foliage and root carbon content based on our allometry (Fig. 4). We also make available our output data for the 1,000 mm year$^{-1}$ precipitation zone southward to 9.5° N latitude with information on location, precipitation, metadata of the Maxar satellite image used, tree crown area, tree wood carbon, tree root carbon and tree leaf carbon.

### Satellite imagery

We used 326,523 Maxar multispectral images from the QuickBird-2, GeoEye-1, WorldView-2 and WorldView-3 satellites collected from 2002 to 2020 from November to March from 9.5° N to 24° N latitude within Universal Transverse Mercator (UTM) zones 28–37 for Africa (Extended Data Table 1a). These images were obtained by NASA through the Next-View License from the National Geospatial-Intelligence Agency. Data were assembled over several years with a focus on later years to achieve a relatively recent and complete wall-to-wall coverage.

When using satellite data from different satellites over several years, with varying sun–target–satellite angles, with varying radiometric calibration of satellite spectral bands and different atmospheric compositions through which the surface is imaged, there are two possibilities for using hundreds of thousands of satellite images together quantitatively. One approach, used extensively in NASA's, NOAA's and the European Space Agency's Earth-viewing satellite programmes, is to quantitatively inter-calibrate radiometrically the satellite channels through time; correct these data for time-dependent atmospheric effects such as aerosols, clouds, haze, smoke, dust and other atmospheric constituent effects and then normalize the viewing perspective to the same sun–target–satellite angle[38]. Another approach is to use the satellite data as collected; assemble training data of trees viewed from different satellites under different sun–target–satellite angles, different times, different atmospheric conditions and use machine learning with high-performance computing to perform the tree mapping at the 50-cm scale. The key to successful machine learning is to account for all the sources of variation within the domain of study in the training data to ensure accurate identification of trees under all circumstances. We included trees viewed substantially off-nadir, trees collected under different aerosol optical thicknesses, trees collected under cirrus cloud conditions, trees viewed in the forward and backward scan directions, trees on sandy soils, trees on clay soils, trees on burn scars, trees in laterite areas and trees in riverine settings. Our training data were collected by one team member and are a carefully selected manual delineation of 89,899 individual trees under a range of atmospheric conditions, viewing perspectives and ecological settings.

All multispectral and panchromatic bands associated with our Maxar images were orthorectified to a common mapping basis. We next pan-sharpened all multispectral bands to the 0.5-m scale with the associated panchromatic band. The absolute locational uncertainty of pixels at the 0.5-m scale from orbit is approximately ±11 m, considering the root-mean-square location errors among the QuickBird-2, GeoEye-1, WorldView-2 and WorldView-3 satellites (Extended Data Table 1). We formed the normalized difference vegetation index (NDVI)[39] from every image in the traditional way from the pan-sharpened red and near-infrared bands. We also associated the panchromatic band with the NDVI band and ensured that the panchromatic and NDVI bands were highly co-registered. The NDVI was used to distinguish tree crowns from non-vegetated background because the images were taken from a period when only woody plants were photosynthetically active in this area[36]. Our training data were labelled on images from the early dry season when only trees have green leaves. Because most semi-arid savannah trees continue to photosynthesize in the early dry season after herbaceous vegetation senesces, green leaf tree crowns are easily mapped because of their higher NDVI values than their senescent herbaceous vegetation surroundings. We substantiate this by analysis of 308 individual trees using NDVI time series with 4-m PlanetScope imagery that emphasized the importance of satellite data from the November, December and January early dry-season months (Extended Data Fig. 1).

We next formed our data into mosaics by applying a set of decision rules, resulting in a collection of 16 × 16-km tiles within each UTM zone from 9.5° N to 24° N latitude for Africa. The initial round of scoring considered percentage cloud cover, sun elevation angle and sensor off-nadir angle: preference was given to imagery that had lower cloud cover, then higher sun elevation angle and finally view angles closest to nadir. In the second round of scoring, selections were assigned priority to favour early dry-season months and off-nadir view angles: preference was given to imagery from November, December and January with off-nadir angle less than ±15°; second to imagery from November to January with off-nadir angle between ±15° and ±30°; third to imagery from February or March with off-nadir angle less than ±15°; and finally to imagery from February or March with off-nadir angle between ±15° and ±30°. Image mosaics were necessary to eliminate multiple counting of trees. We formed mosaics using 94,502 images for tree segmentation, with 94% of these being from November, December and January. Ninety percent of our selected mosaic imagery was within ±15° of nadir, 87% were acquired between 2010 and 2020 and 94% were from the early dry season (Extended Data Fig. 7). A summary of month, year, solar elevation and off-nadir angle by UTM zone can be found in Supplemental Information Fig. 1.

Possible obscuration of the surface by clouds totalled 4.1% of our input mosaic data area and aerosol optical depth >0.6 at 470-nm (ref. [40]) areas totalled 3.4% of our input data. However, we mapped 691,477,772 trees in our possible cloud-cover-affected and aerosol-affected areas, indicating that cloud and aerosol effects were lower than these numbers. In addition, 0.9% of our input data did not process. We include a data layer in our viewer for these three conditions.

### Mapping tree crowns with deep learning

We used convolutional neural network models developed by a previous study[1]. The models were trained with manually delineated and annotated 89,899 individual trees along a north–south gradient from 0 to 1,000 mm year$^{-1}$ rainfall[1]. Only features that showed a distinct crown area and associated shadow were included, which excluded small bushes, grass tussocks, rocks and other features that might have green leaves or cast a shadow from our classification. All training data and model training was done in UTM zones 28 and 29. Because tree floristic diversity in the 0–1,000 mm year$^{-1}$ zone of our study is highly similar from the Atlantic Ocean to the Red Sea across Africa[41–43], we added no further training data as our study moved further eastward. We used state-of-the-art deep learning to segment trees crowns at the 50-cm scale[1]. We used two different models based on a U-Net architecture, one for lower-rainfall desert regions with <150 mm year$^{-1}$ precipitation and one for regions with average annual precipitation >150 mm year$^{-1}$. Details about the network architecture, training process and hyperparameter choices can be found in ref. [1]. Previous evaluation showed that early dry-season images performed better than late dry-season images, which was a limitation of our previous study. We reduced this error by using early dry-season images with only 6% of our area being

covered by images from February and March. The models were also designed to separate clumped trees by highlighting spaces between different crowns during the learning process, similar to a strategy for separating touching cells in microscopic imagery[22].

## Allometry

Very-high-resolution satellite images and deep learning have achieved mapping of individual trees over large areas[1]. Each tree is georeferenced in the satellite data and defined by crown area. The challenge was to develop allometric equations for foliage, wood and root dry masses or carbon based on crown area regardless of species. This was met by reanalysing existing Sahelian and Sudanian woody plant data from destructive sampling. Overall, the seasonal maximum foliage, wood and root dry masses were measured on 900, 698 and 26 trees or shrubs from 27, 26 and 5 species, respectively, for which crown area was also measured. Several allometric regression models tested for foliage, wood or root masses are power functions and independent of species. All the regression outputs were inter-compared for fit indicators, by systematic estimates of prediction uncertainty and by root-to-wood ratios and foliage-to-wood ratios over the range of crown areas. This resulted in a set of ordinary least squares log–log equations with crown area as the independent variable. The Sahelian and Sudanian allometry equations were also compared with published allometry equations for tropical trees, primarily from more humid tropics, which are generally based on stem diameter, tree height and wood density. Our allometric predictions are within the range of other allometry predictions, reinforcing the confidence in their use beyond the Sahelian and Sudanian domains into sub-humid savannahs for discrete trees[19].

On the basis of ref. [19], we predicted the wood (w), foliage (f) and root (r) dry mass as functions of the crown area ($A$) of a single tree as:

$$\text{mass}_w(A) = 3.9448 \times A^{1.1068} \ (N_w = 698)$$
$$\text{mass}_f(A) = 0.2693 \times A^{0.9441} \ (N_f = 900)$$
$$\text{mass}_r(A) = 0.8339 \times A^{1.1730} \ (N_r = 26)$$

The tree mass components of wood, leaves and roots were combined to predict the total mass($A$) in kg of a tree from its crown area $A$ in m$^2$:

$$\text{mass}(A) = \text{mass}_w(A) + \text{mass}_f(A) + \text{mass}_r(A)$$

As in ref. [1], a crown area of size $A > 200$ m$^2$ was split into $\lfloor A/100 \rfloor$ areas of size 100 m$^2$ and one area with the remaining m$^2$ if necessary. We converted dry mass to carbon by multiplying with a factor of 0.47 (ref. [44]).

## Uncertainty analysis

We evaluated the uncertainty of our tree crown area mapping and carbon estimation in two ways. First, we quantified our tree crown mapping omission and commission errors by inspecting randomly selected areas from UTM zones 28–37, validating that our neural network generalized over UTM zones consistently (Extended Data Fig. 8).

Second, we quantified the relative error of our tree crown area estimation. We consider the uncertainty $\Delta_x$ of a quantity $x$ and the corresponding relative uncertainty $\delta_x$ defined by the absolute and relative error, respectively[45]. To assess the relative error in crown area estimation resulting from errors by the neural network, we considered external validation data from ref. [1], which were not used in the model-building process. We considered expert-labelled tree crowns as well as the predicted tree crowns from 78 plots of 256 × 256 pixels. The hand-labelled set contained 5,925 trees and the system delineated 5,915 trees. The total hand-labelled tree crown area was 118,327 m$^2$ and the neural network predicted 121,898 m$^2$. This gave a relative error in crown area mapping of $\delta_\text{area} = 3.3\%$. We matched expert-labelled and predicted tree crowns and computed the root-mean-square error (RMSE) per tree, taking overlapping areas and missed trees into account (see Extended Data Fig. 8). We estimated the allometric uncertainty ($\delta_\text{allometric}$) using the data

from ref. [19] (see below). The two relative errors $\delta_\text{area}$ and $\delta_\text{allometric}$ were combined to an overall uncertainty estimate for the carbon prediction of ±19.8% (see below).

## Omission and commission errors

We evaluated our tree crown mapping accuracy by analysis of 1,028 randomly selected 512 × 256-pixel areas over the 9.5° N to 24° N latitude within UTM zones 28–37. Because the drier 60% of our study area only contains 1% of the 9,947,310,221 trees we mapped in the 0–1,000 mm year$^{-1}$ rainfall zone, we applied an 80% bias for selecting evaluation areas above the 200 mm year$^{-1}$ precipitation line[46], as >98% of tree identifications were above the 200 mm year$^{-1}$ precipitation isoline. Identified tree polygons were further categorized into tree crown area classes from 0–15 m$^2$, 15–50 m$^2$, 50–200 m$^2$ and >200 m$^2$, with a total of 50,570 trees evaluated. Although a previous study reported greatest uncertainty in both the smallest and largest area classes[1], our more expansive work found the greatest uncertainty in our smallest tree class. We excluded from evaluation any tiles that had annual precipitation[46] >1,000 mm year$^{-1}$ and all areas that were devoid of vegetation, leaving us with 850 areas.

Seven members of our team evaluated the accuracy in terms of commission and omission by tree crown area classes for the 850 areas. Input data provided for every area were the NDVI layer, the panchromatic shadow layer and the neural net mapping results in each of the four crown area classes. Ancillary data available to evaluators included the centre coordinates for comparison with Google Earth data, the Funk et al.[46] rainfall, the acquisition date of the area evaluated and the viewing perspective.

We identified areas wrongly classified as tree crowns (commission errors), missed trees (omission errors) and crown areas corresponding to clumped trees (Extended Data Fig. 8). Clumped trees were most common for >200 m$^2$ tree crown area. They were rare in the 3–15 m$^2$ and 15–50 m$^2$ tree classes, which comprise 88% of our tree crowns. In the 850 patches, the number of trees ranged from one tree to 326 trees, with a total of 50,570 trees evaluated and 3,765 errors identified. Overall, the commission and omission error rates were 4.9% and 2.7%, respectively, a net uncertainty of 2.2%.

## Allometric uncertainty estimation

The prediction of tree carbon from the crown area for a single tree based on crown area alone is inherently uncertain[47,48]. As the allometric equations are based on three different datasets, we compute their uncertainties independently, combine them and put them in relation to the total carbon measured in the three datasets.

The allometric equations were established using an optimal least-squares fit of an affine linear model predicting the logarithmic carbon from the logarithmic tree crown area[19]. To estimate the uncertainty of the allometric equations, we repeated the fitting using random subsampling. The datasets were randomly split into training data (80%) for fitting the allometric equations and validation data (20%) for assessing the uncertainty. For example, from the root measurements, $(A_1, y_1), \ldots, (A_{N_r}, y_{N_r})$, we compute $\mu_r = \frac{1}{N_r} \sum_{i=1}^{N_r} y_i$ and $\hat{\mu}_r = \frac{1}{N_r} \sum_{i=1}^{N_r} \text{mass}_r(A_i)$. The corresponding error is $\Delta_r = |\mu_r - \hat{\mu}_r|$.

Because the total carbon for a tree with a certain crown area is the sum of the three carbon components, we add the absolute uncertainties assuming independence[45].

$$\Delta_\text{allometric} \simeq \sqrt{\Delta_f^2 + \Delta_w^2 + \Delta_r^2}$$

and compute the relative uncertainty as $\delta_\text{allometric} = \frac{\Delta_\text{allometric}}{\mu_\text{mass}}$, in which the average mass $\mu_\text{mass}$ is given by the sum of the averages for wood ($\mu_w$), leaves ($\mu_f$) and root ($\mu_r$). This process was repeated ten times, resulting in a mean relative uncertainty of

$$\overline{\delta}_\text{allometric} = 19.5\%.$$

## Total carbon uncertainty

We combine the uncertainties from the neural net mapping and our allometric equations, which can be viewed as considering $(1 + A) \cdot (1 + B)$ with $A$ and $B$ being random variables with standard deviations $\delta_{area}$ and $\delta_{allometric}$. Neglecting higher-order and interaction terms, we combine the two sources of uncertainty to $\delta \simeq \sqrt{\delta_{area}^2 + \overline{\delta}_{allometric}^2}$, resulting in an uncertainty in total tree carbon for our study of ±19.8%. See also Extended Data Fig. 9 for the RMSEs of our predicted crown areas calculated on external validation data from ref. [1], binned on the basis of the 50th quantiles of the hand-labelled crown areas and converted also into carbon. Extended Data Fig. 10 is a flow diagram summarizing our methods.

## Our viewer

Visualizing our large tree-mapping dataset in an interactive format was essential for quality-control purposes, exploration of the data and hypothesis creation. Creating a web-based viewer serves the purpose of being the initial point of interaction with our dataset for fellow researchers, local stakeholders or the general public. The visualization of more than 10 billion trees in a web browser required maintaining performance, interactivity and individual metadata for each polygon. Users should be able to zoom in to any area within the dataset to view individual tree polygons and query their statistics while at the same time accurately depicting the overall trends of the dataset at lower zoom levels. The visualization also needed to clearly denote where data were missing or possibly affected by clouds or aerosols. Finally, the extent and origin of the source imagery, its acquisition date and a preview of the imagery needed to be available. To accomplish these goals, a vector-tile-based approach was taken, with the data visualized in a Mapbox GL JS map within a React web application. To create vector tiles covering the entire study area, we developed a data-processing pipeline using high-performance computing resources to transform the data into compatible formats, as well as to package, optimize and combine the vector tiles themselves.

We used two tracks to store and visualize the results of this study on the web: vector polygon data and generalized rasters representing tree crown density. At the native spatial resolution of 50 cm, the map shows the full-resolution tree polygon dataset. At lower-spatial-resolution zoom levels, rasterized representations of tree density are shown. Visualizing generalized rasters in place of vector polygons improves performance substantially. As users zoom in to higher spatial resolutions, the raster layer fades away and is replaced by the full-resolution polygon layer. Once zoomed far enough to resolve individual polygons, users can click to select a polygon to show a map overlay containing various properties of the tree, as well as the date on which the source imagery was acquired and a link to preview the source imagery.

**Rainfall data.** We used the rainfall data of Funk et al. to estimate annual rainfall at 5.6-m grids[46]. We averaged the available data from 1982 to 2017 and extracted the mean annual rainfall for each mapped tree and bilinearly interpolated it to $100 \times 100$-m resolution. The rainfall data were also used to classify the study area into mean annual precipitation zones: hyper-arid from 0–150 mm year$^{-1}$, arid from 150–300 mm year$^{-1}$, semi-arid from 300–600 mm year$^{-1}$ and sub-humid from 600–1,000 mm year$^{-1}$ zones. The rainfall data are found at https://data.chc.ucsb.edu/products/CHIRPS-2.0/africa_monthly/ (ref. [46]).

## Data availability

The viewer can be accessed at https://trees.pgc.umn.edu/app. The Funk et al. rainfall data[46] are freely available at https://data.chc.ucsb.edu/products/CHIRPS-2.0/africa_monthly/. Commercial very-high-resolution satellite images were acquired through NASA under the NextView Imagery End User License Agreement. The copyright remains at Maxar, Inc. and redistribution is not possible. However, the derived products produced by this study are publicly available at the Oak Ridge National Laboratory's Distributed Active Archive Center: https://doi.org/10.3334/ORNLDAAC/2117. Please contact C.T., M.B. or P.H. for more specific requests. A detailed description of our processed data for the 95,402 selected mosaic images, including output data, specific cutlines affected by aerosol optical depth and cloud cover; data distributions for year, month, solar azimuth angle and off-nadir angle for each UTM zone segment in our study, can also be found at https://doi.org/10.3334/ORNLDAAC/2117.

## Code availability

The tree detection framework based on U-Net is publicly available at https://zenodo.org/record/3978185. Please contact A.K., C.I., M.B. or J.M. for support and more information.

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

**Acknowledgements** We thank Maxar, Inc. for providing commercial satellite data through the NextView Imagery End User License Agreement. M.B. and F.R. were supported by the European Research Council (ERC) under the European Union's Horizon 2020 research and innovation programme (grant agreement no. 947757 TOFDRY) and a DFF Sapere Aude grant (no. 9064–00049B). A.K., R.F. and C.I. acknowledge support by the Villum Foundation through the project Deep Learning and Remote Sensing for Unlocking Global Ecosystem Resource Dynamics (DeReEco, grant no. 34306). C.I. acknowledges support by the Pioneer Centre for Artificial Intelligence, DNRF grant number P1. NASA supported this work through the Commercial SmallSat Data Acquisition (CDSA) Program. We thank NASA's K. Murphy and A. Hall for funding this work; T. Lee and D. Duffy of NASA for supporting our use of Maxar data; B. Kramer, B. Bode and the entire Blue Waters staff who enabled our extensive use of high-performance computing to map 9,947,310,221 trees and convert them into wood, foliage and root carbon.

**Author contributions** C.T., M.B., P.H. and A.K. contributed equally to the paper and should be considered as the first authors. C.T. and M.B. coordinated the study and drafted the manuscript, with support by K.R., R.F. and P.H. P.H., P.S. and B.-A.I. collected, assembled and developed the allometric equations. Satellite data were prepared by J.S., S. Sinno, J.M., C.P. and Y.F. J.P. and J.M. directed the evaluation, which was conducted by K.M., E.R., D.M., A.M., A.K., J.S. and C.T. The tree detection was processed on the Blue Waters supercomputer by J.M., J.S. and S. Sinno. Neural network implementation was provided by A.K. and C.I. Ecosystem models were run and analysed by B.P. and P.C. LVOD data were prepared by J.-P.W. and M.B. Data were analysed by J.P., E.R., E.G., R.F., J.M., D.M., A.M., A.K., K.M., J.S., M.B., F.R., C.T., C.I., R.K., R.M., S. Saatchi and Y.F. Phenology was provided by Y.F., P.H. and L.K. The viewer was developed by P.M. and S.L.

**Competing interests** The authors declare no competing financial interests.

**Additional information**
**Correspondence and requests for materials** should be addressed to Compton Tucker, Martin Brandt or Pierre Hiernaux.

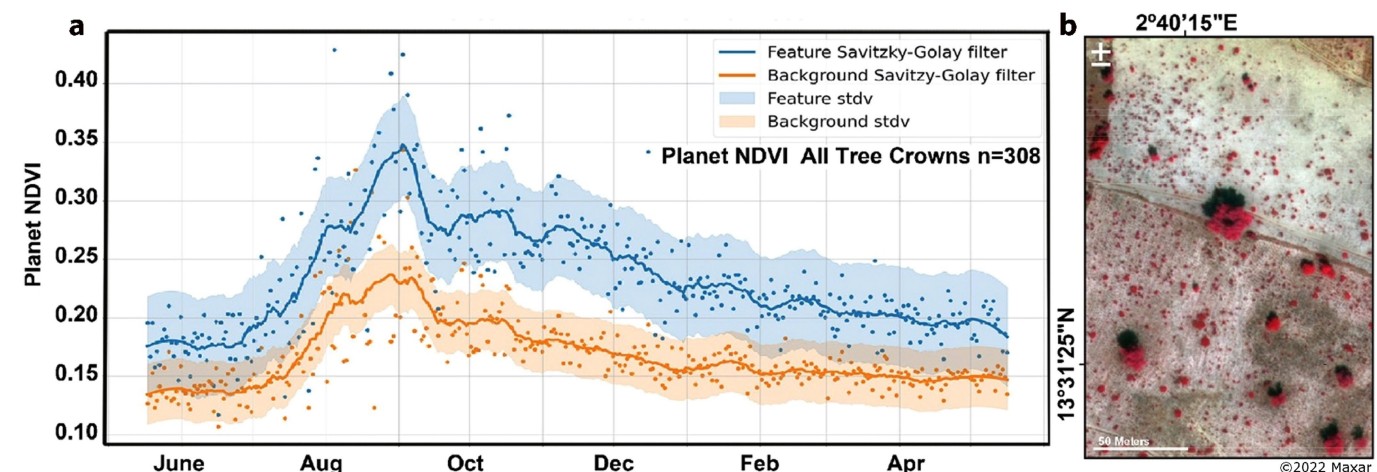

**Extended Data Fig. 1 | Planet NDVI time series for 308 trees contrasted with the background NDVI from the same area. a**, Note the separation between the tree NDVI values and the background NDVI for the dry-season months of November to March. **b**, False-colour tree crown NDVI image from WorldView-3 at 30-cm spatial resolution showing green leaf vegetation in red colours. Individual trees are evident, with green leaf tree crowns with associated shadows. Our mapping of trees with machine learning is based on trees defined as having a tree crown of at least 3.0 m² with an associated shadow. Areas of low vegetation do not cast shadows sufficiently to be classified as trees. The areas in **a** and **b** are close to 13° 31′ N × 2° 40′ E.

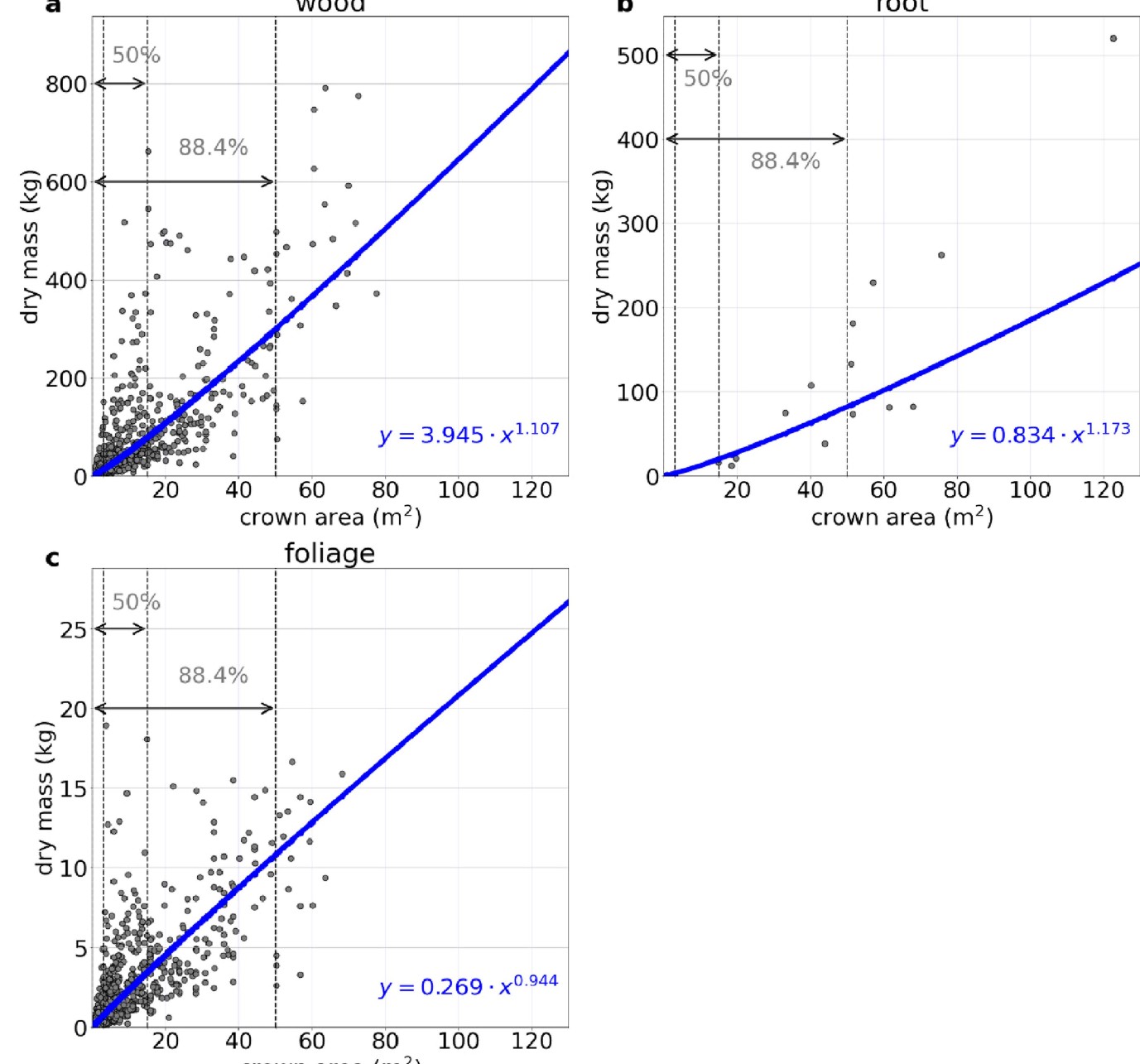

**Extended Data Fig. 2 | Allometric equations for converting tree crown areas into dry wood mass, dry root mass and dry foliage mass. a**, Allometric equation based on tree crown area to predict wood mass was established from 698 Sahelian and Sudanian woody plants of 27 species. The power model was fitted using log–log regression. The stored carbon is estimated from dry mass by multiplying with a factor of 0.47. Data were collected in the 0–800 mm year$^{-1}$ long-term precipitation zone (see ref. [19]). The plot also shows the cumulative percentage up to 15 m$^2$ and 50 m$^2$ crown area (50% and 88.4%, respectively) of the predicted trees in our study (95% of which had crown areas less than 78 m$^2$). **b,c**, Same as in **a** but for foliage and root mass. The foliage and root allometric equations were established from 900 trees of 26 species and 26 trees of five species, respectively.

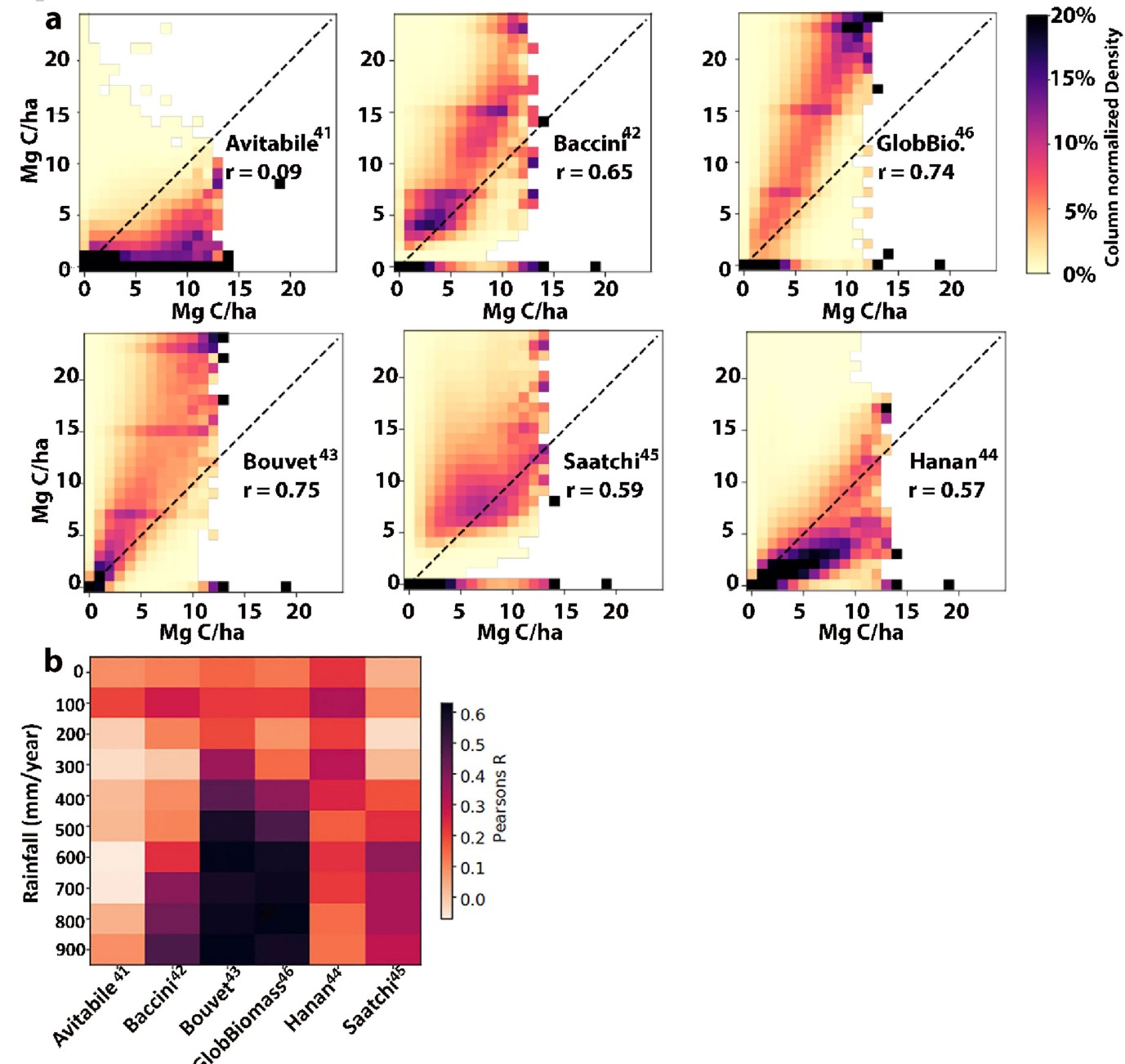

**Extended Data Fig. 3 | Comparison between aboveground carbon-density maps and our data derived from 9,947,310,221 trees. a**, Scatter plots showing the spatial agreement at the pixel level with all datasets aggregated to 1 × 1-km grids, with our data plotted on the *x* axes. **b**, We correlated all 1 × 1-km pixels for each rainfall zone, defined by 100 mm year⁻¹ steps, between our aboveground carbon density (foliage + wood) with the estimates and current state-of-the-art aboveground carbon-density maps[6–11] and our data. See also Fig. 2a.

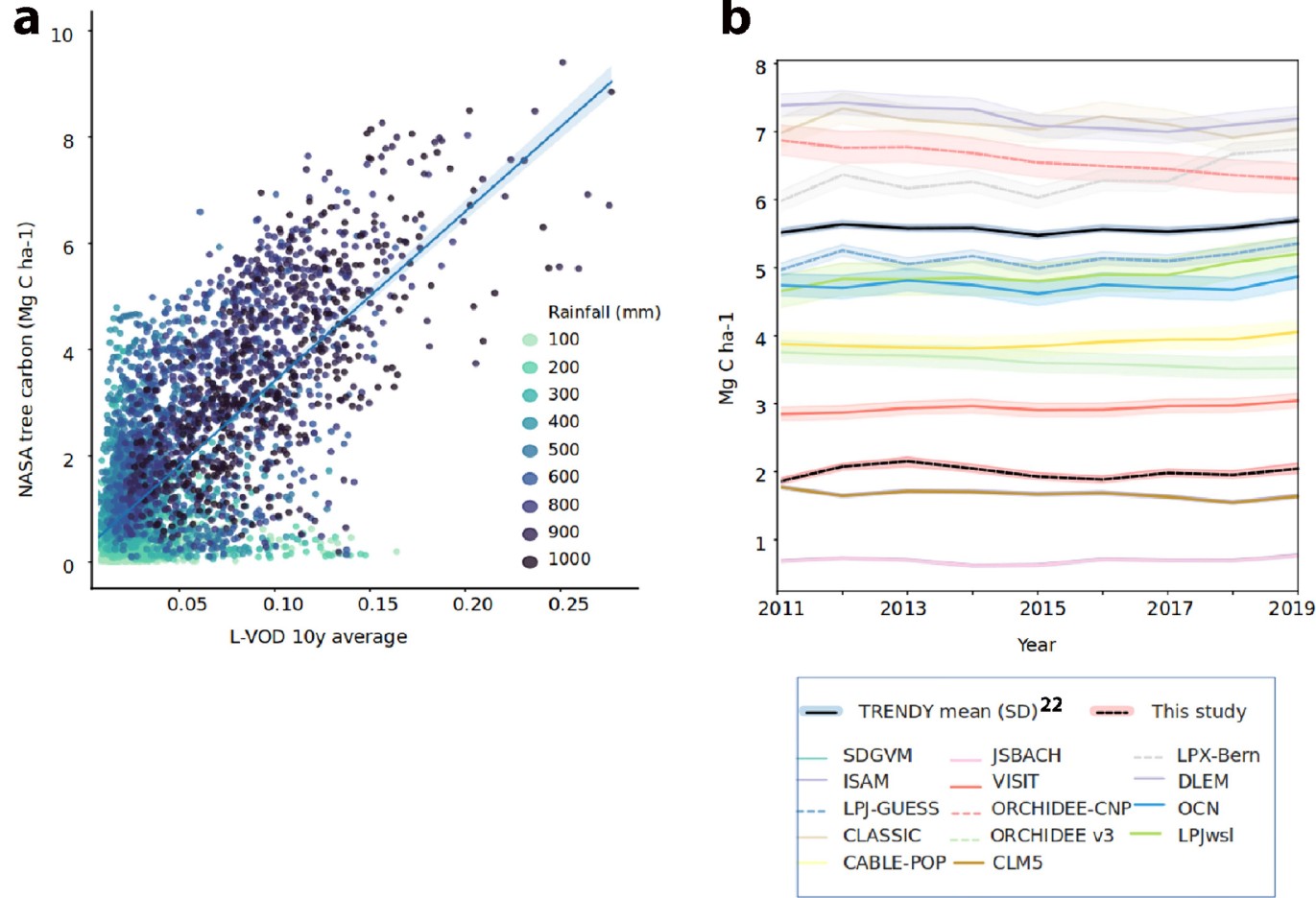

**Extended Data Fig. 4 | Temporal changes in carbon density. a**, Scatter plot between the passive microwave L-VOD[36] and our carbon density (wood + foliage) aggregated to 25 × 25-km grids. **b**, The linear relationship seen in **a** was used to convert annual L-VOD to the unit carbon density. L-VOD aboveground woody carbon density as well as TRENDY models[23] values (all vegetation carbon) were averaged over the study area for the 0–1,000 mm year$^{-1}$ rainfall zone for the period 2011–2019. Correlating L-VOD from the dry season, to avoid the complication of herbaceous vegetation, with our carbon-density map aggregated to 25 × 25-km resolution showed a moderately high level of agreement ($r = 0.72$); however, the strong scattering especially in low-rainfall areas also showed that the uncertainty was high, impeding the use of L-VOD[37] for local applications in arid areas. Nevertheless, the linear relationship was used to convert L-VOD to the unit carbon density to derive temporal dynamics in carbon density, which showed stable woody carbon stocks during 2010–2019 (about 2.0 Mg C year$^{-1}$) for this region, without clear trend or inter-annual variations, suggesting that none of droughts, deforestation or restoration had a measurable impact on carbon stocks over the past decade. TRENDY models showed a variety of responses but the ensemble showed a similar behaviour as L-VOD, although when herbaceous vegetation and belowground biomass were included, a variety of different magnitudes resulted.

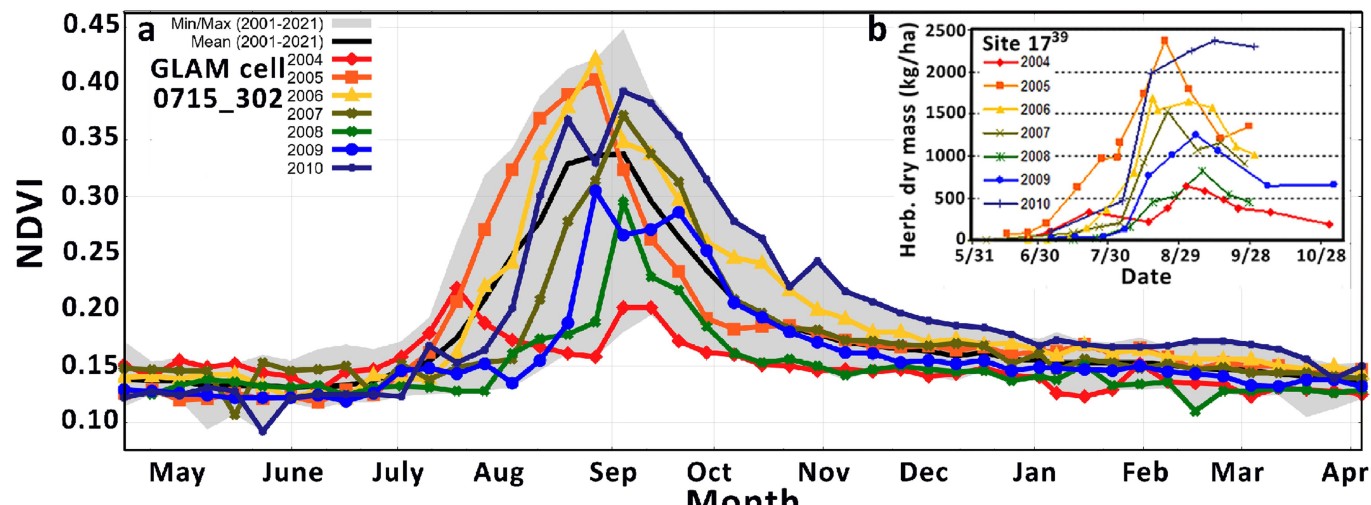

**Extended Data Fig. 5 | Seasonal comparison between MODIS NDVI from a 0.25° × 0.25° area and field data collections of dry herbaceous biomass production in kg ha⁻¹ near Agoufou, Mali within the MODIS area. a**, The MODIS 8-day time-step data showed the NDVI maximum and minimum range from 2000 to 2021 in the grey-coloured portion, with MODIS NDVI time series by years for 2004 to 2010 and the average MODIS NDVI from 2001 to 2021. **b**, Aboveground herbaceous dry mass biweekly in situ measurements for a rangeland field site in Mali[49] centred at 15.4625° N by 1.4886° W. These showed high inter-annual and intra-annual herbaceous dry mass variability, sharp increases in herbaceous dry mass during the wet season and rapid decreases in early dry season at the few-metres scale. The MODIS NDVI data showed similar temporal trends to the herbaceous dry-mass variations in **b** for 2004–2010 and put these into context of the 2000–2021 MODIS record for a 0.25° × 0.25° area. Trees was roughly 3% of the total vegetation cover in this area and thus herbaceous vegetation dominates. The species composition in **b** was dominated by annual grasses, such as *Aristida mutabilis*, *Cenchrus biflorus* and *Brachiaria xantholeuca*, and annual dicotyledons, such as *Zornia glochidiata* and *Tribulus terrestris*. The data in **b** were from a 1 × 1-km location and were selected to be representative of the area. The MODIS NDVI data in **a** are available for all users, with instructions for use for the MODIS record from 2000 to the present[50]. Our tree crown data provide the means to separate primary production into herbaceous and tree fractions for semi-arid areas and will improve carbon residence understanding in areas of mixed tree and herbaceous vegetation.

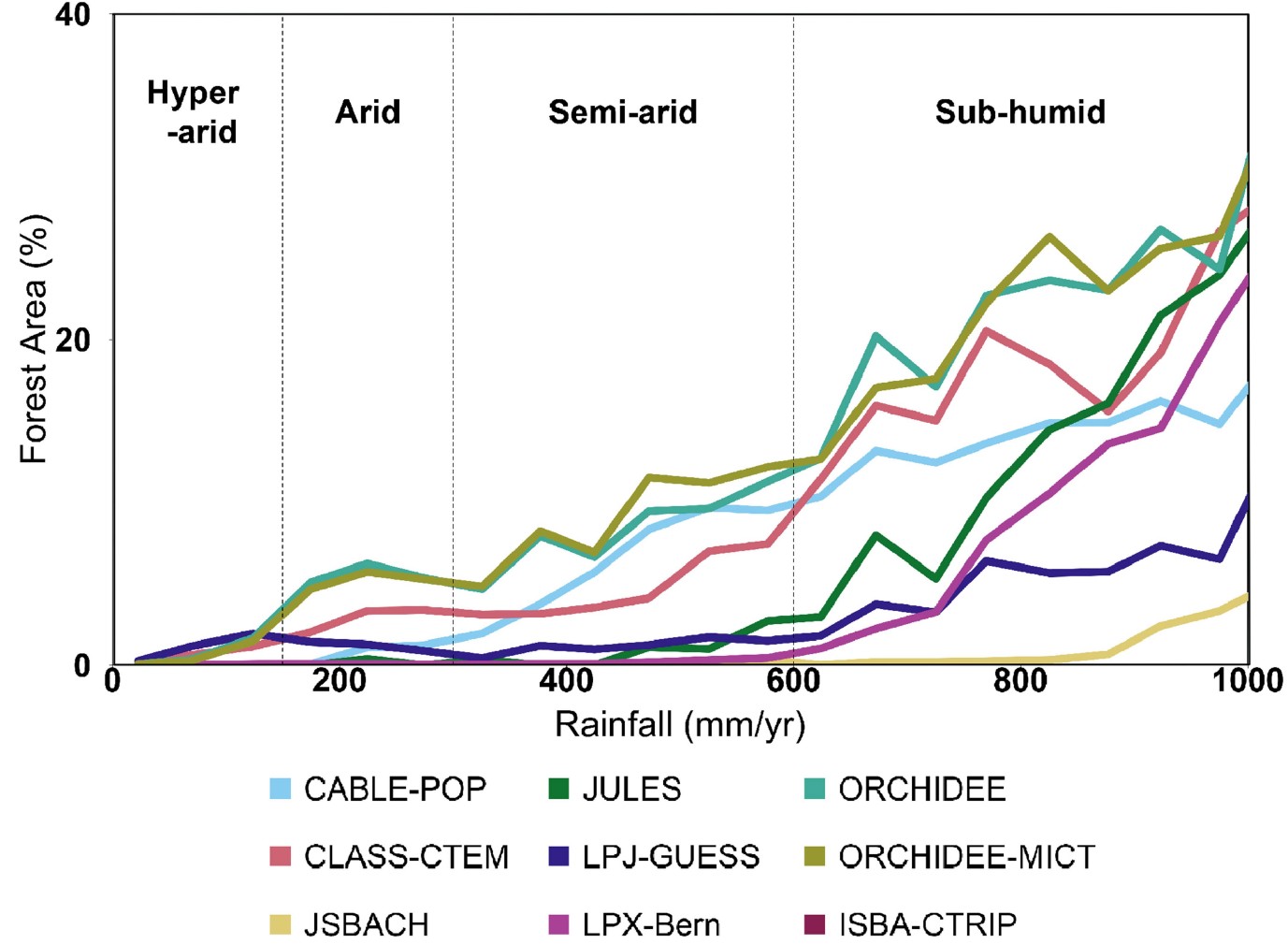

**Extended Data Fig. 6 | Forest fractions of different TRENDY[23] models.** The figure shows the percentage of forest areas assumed by each model along the rainfall gradient. Woody biomass in most models mainly comes from predefined forest areas and consequently results in a high degree of variation in the semi-arid area of our study. This is an example of the utility of our tree-mapping results for more accurate depiction of semi-arid trees for numerical simulation modelling.

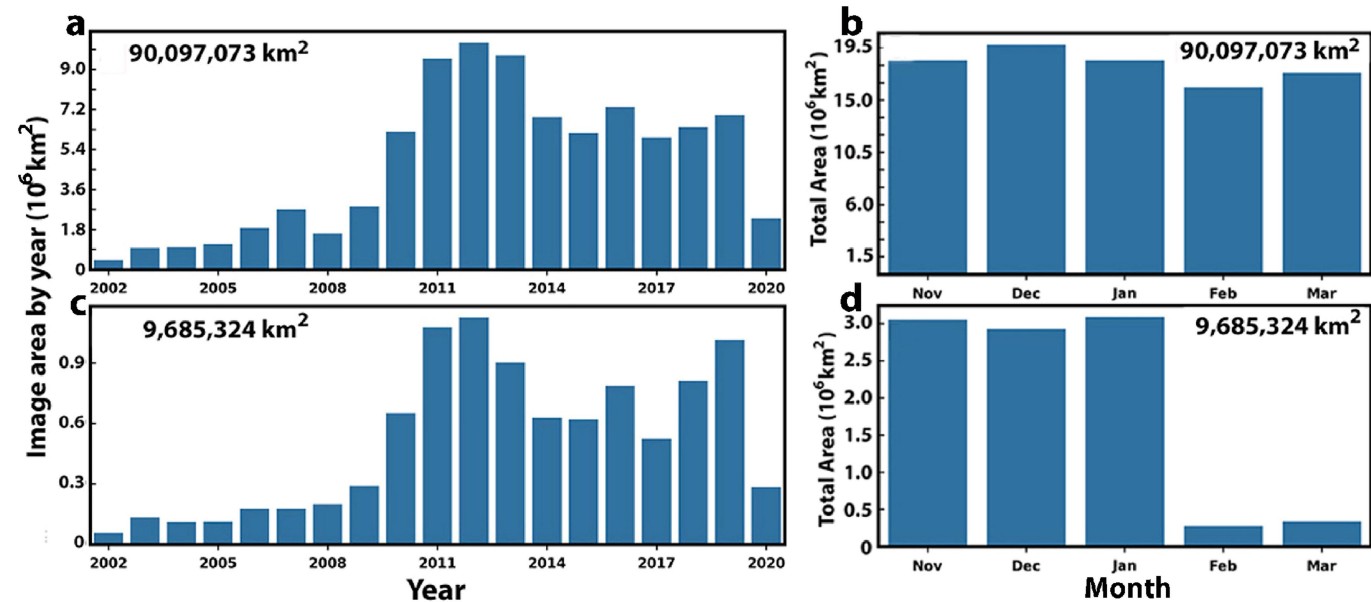

**Extended Data Fig. 7 | Candidate and selected satellite images for our analysis. a**, 326,523 Maxar images covering 90,097,073 km² from November to March were available for our study and were acquired from 2002 to 2020. **b**, The distribution by month of all satellite data in **a**. The selected imagery for processing (see Methods) by year totalled 94,502 images that covered 9,685,324 km² with 87% of the satellite data from 2010 to 2020 (**c**) and 94% of the selected images were from the early dry-season months of November, December and January (**d**). We had a 9.3:1 area ratio of available imagery to selected input data for analysis. See also Extended Data Table 1a.

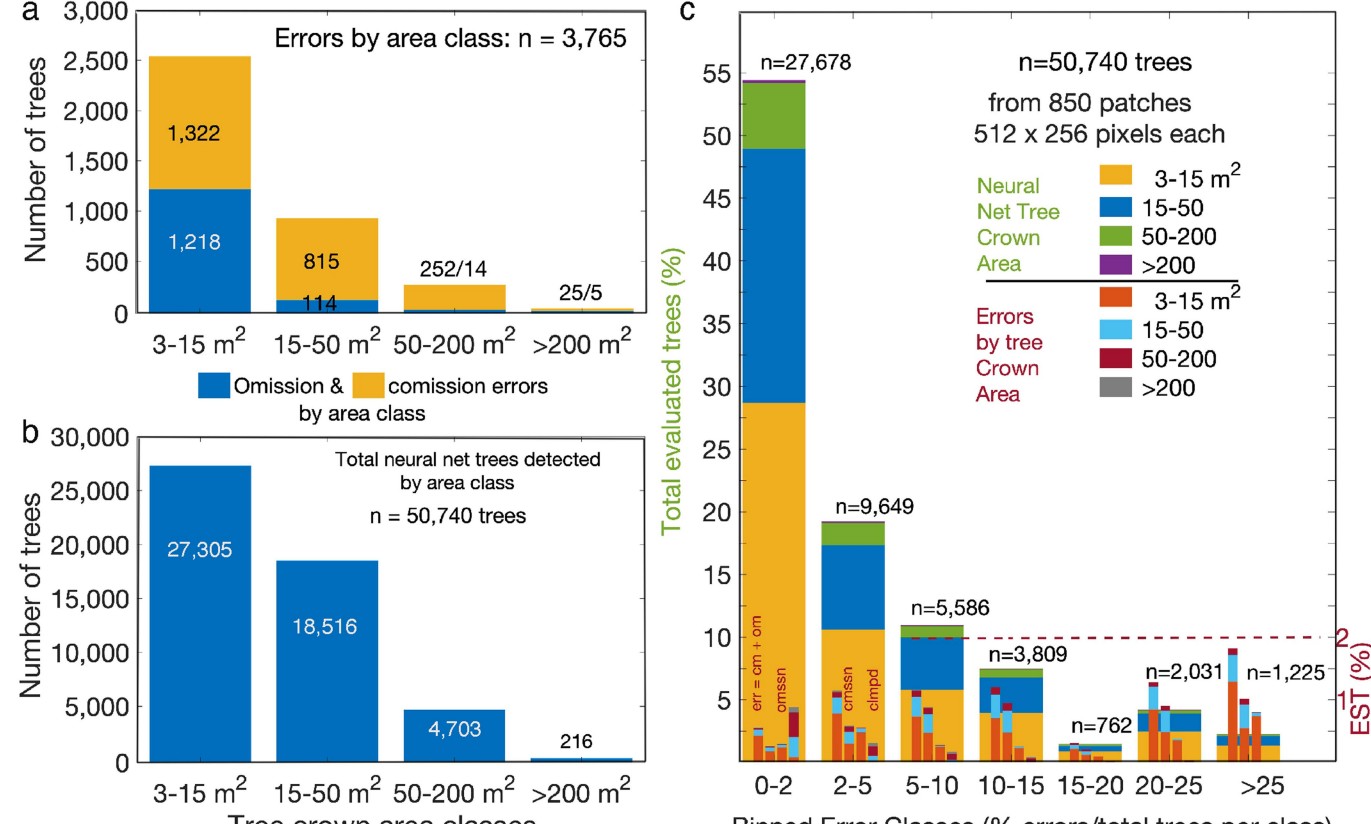

**Extended Data Fig. 8 | Evaluation of the tree crown mapping omission and commission errors.** The performance of the tree crown predictions was evaluated for 1,028 randomly selected 512 × 256-pixel areas from UTM zones 28–37, with approximately 100 areas for each UTM zone. The patches were extracted with an 80% bias towards precipitation[46] above 200 mm year$^{-1}$, as most tree identifications are above the 200 mm year$^{-1}$ isoline. 178 areas were excluded from evaluation because the rainfall was >1,000 mm year$^{-1}$ or the areas were devoid of trees. A total of 50,740 trees were evaluated. **a**, Class breakdown of our 3,765 omission and commission errors. **b**, Class numbers of all the trees evaluated. **c**, Summarizes the number of trees, errors of commission and omission and the number of trees by error classes. The highest percentage of binned-error classes, from 15% to >25%, occurred in areas with few trees and resulted from mixed tree and bush confusion for <8% of all trees evaluated. By contrast, the lowest binned-error classes, from 0% to 15%, had 92% of all trees evaluated. In the 850 patches, the number of trees ranged from one tree to 326 trees, with a total of 50,740 trees evaluated and 3,765 errors identified. Overall, the commission and omission error rates were 4.9% and 2.7%, respectively, a net uncertainty of 2.2%.

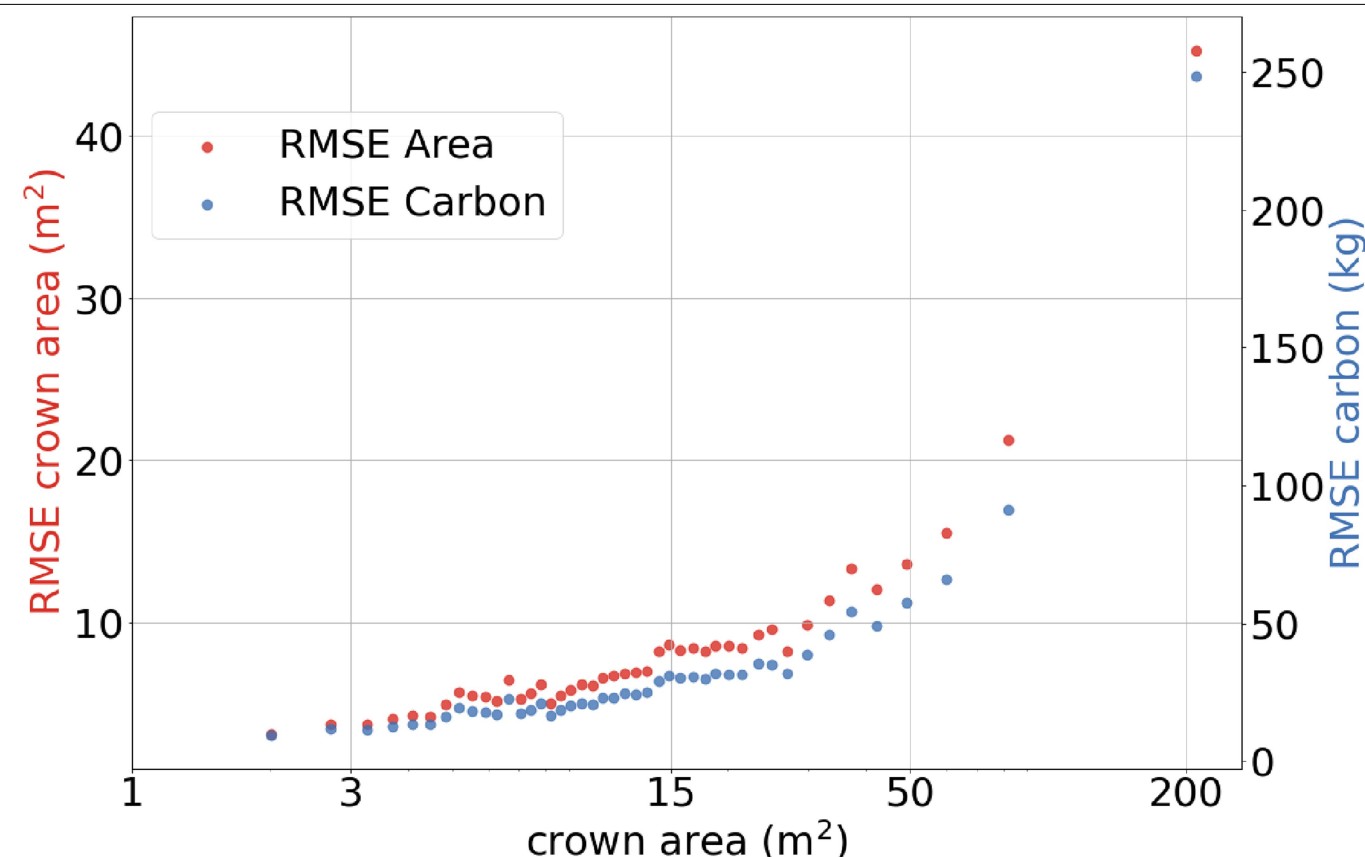

**Extended Data Fig. 9 | Tree crown and carbon errors.** The RMSEs of our predicted crown areas calculated on external validation data from ref.[1], binned on the basis of the 50th quantiles of the hand-labelled crown areas. In 78 plots of 256 × 256 pixels, the hand-labelled set contained 5,925 trees and the system delineated 5,915 trees. These crown areas were matched using inner spatial join. Multiple overlapping hand-labelled or predicted tree crown areas were merged into multi-polygons before calculating the RMSE. The crown areas of missed tress counted as errors. For calculating the corresponding RMSE of predicted carbon, we relied on the allometric equations given in Extended Data Fig. 2a–c. The abscissa has a logarithmic scale and 95% of our 9.9 billion tree crowns had crown areas <78 m² (Fig. 3b).

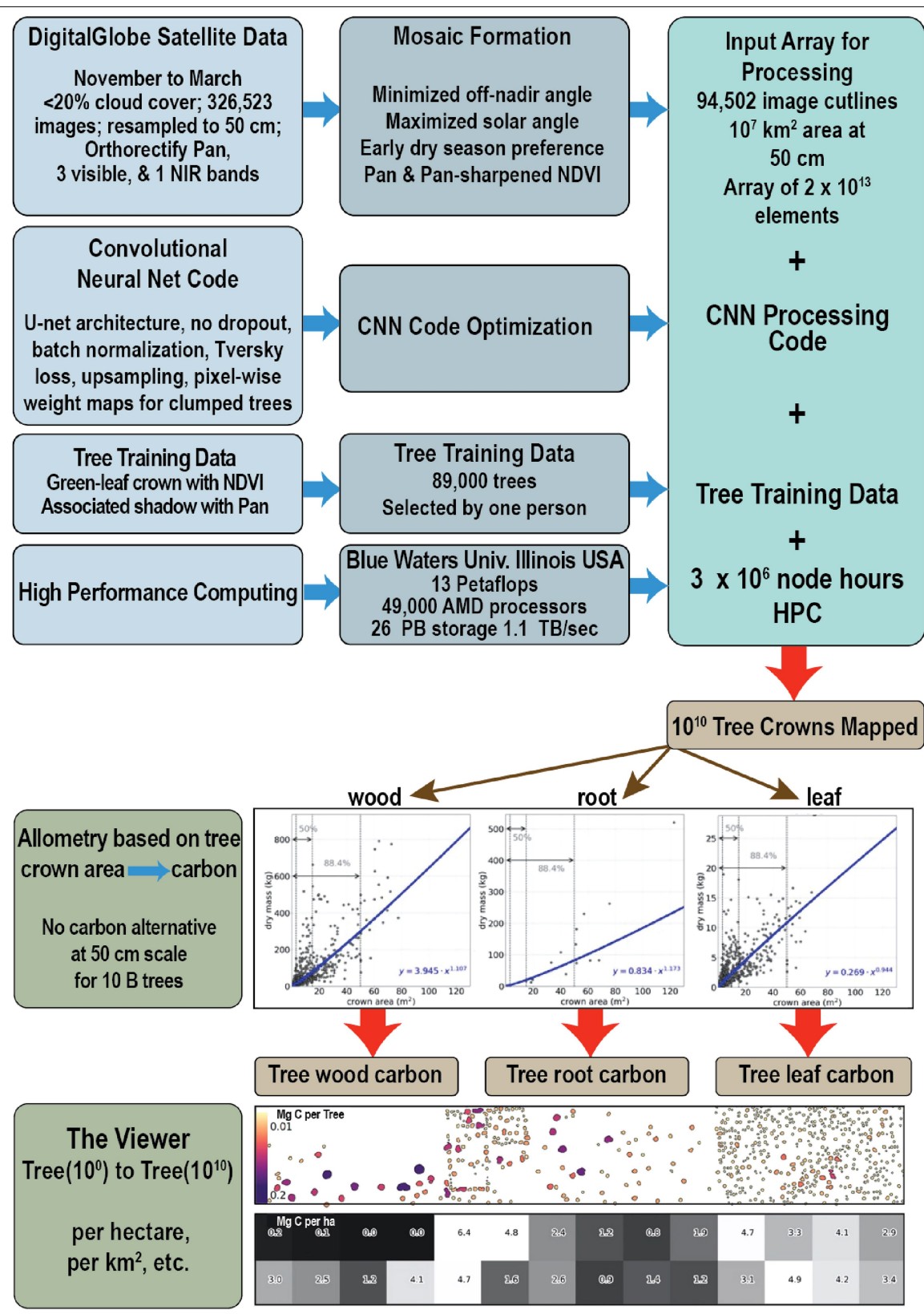

**Extended Data Fig. 10 | Flow chart of the different components of our paper.**
We show the satellite data used, how these data were organized for processing with our machine learning software and the requisite training data, and how the resulting segmentation of ten-billion-tree crown area resulted. We then show the conversion of tree crown area in tree wood, root and leaf carbon at the tree level for ten billion trees. Last, we show the use of our viewer to enable use of the data we produced, from tree (1) to tree ($10^{10}$).

**Extended Data Table 1 | The November to March Maxar satellite images considered and the number selected for our analysis by UTM zone**

**a.**

| UTM Zone | Total Images | Selected for Mosaic | GeoEye-1 | QuickBird-2 | WorldView-2 | WorldView-3 |
|---|---|---|---|---|---|---|
| 32628 | 42,388 | 8,026 | 1,557 | 1,501 | 3,897 | 1,071 |
| 32629 | 29,244 | 9,883 | 2,208 | 2,198 | 4,334 | 1,143 |
| 32630 | 32,671 | 10,155 | 2,518 | 2,130 | 4,401 | 1,106 |
| 32631 | 34,611 | 10,158 | 2,384 | 2,049 | 4,554 | 1,171 |
| 32632 | 32,210 | 9,971 | 2,397 | 1,876 | 4,651 | 1,047 |
| 32633 | 34,918 | 10,097 | 2,688 | 2,082 | 4,310 | 1,017 |
| 32634 | 31,555 | 10,144 | 2,353 | 2,288 | 4,637 | 866 |
| 32635 | 36,663 | 9,975 | 2,269 | 2,099 | 4,595 | 1,012 |
| 32626 | 36,934 | 10,156 | 2,003 | 1,932 | 5,061 | 1,160 |
| 32637 | 15,329 | 5,937 | 1,079 | 1,353 | 2,742 | 763 |
| ALL | 326,523 | 94,502 | 21,456 | 19,508 | 43,182 | 10,356 |

**b.**

| Quickbird-2 | GeoEye-1 | WorldView-2 | WorldView-3 |
|---|---|---|---|
| Oct. 2001-Feb. 2015 | Nov. 2008 | Oct. 2009 | Aug. 2014 |
| 15 km & 55 cm | 15.3 km & 41 cm | 16.4 km & 46 cm | 13.1 km & 31 cm |
| Panchromatic, Visible, & Near IR | Panchromatic, Visible, & Near IR | Panchromatic, Visible, & Near IR | Panchromatic, Visible, Near IR, & SWIR |
| 23 m CE90 | 4.0 m CE90 | 5.0 m CE90 | 3.7 m CE90 |
| 10.8 m RMSE | 2.7 m RMSE | 3.0 m RMSE | 2.5 m RMSE |

**a**, We started with 326,523 candidate images and selected 94,502 for processing for the area of 9.5°N to 24°N latitude from the Atlantic Ocean to the Red Sea for the months of November to March. 87% of data selected for processing were acquired from 2010 to 2020 and 94% of the selected satellite images were from November to January. Each image had a panchromatic and NDVI component that was used to identify trees with canopy area 3 m$^2$ or greater in the early dry season. **b**, Specific satellite information for the four Maxar satellites used in our study includes the period of operation or the launch date, the nadir swath width, the nadir panchromatic spatial resolution and the spectral regions available. The relative geolocation accuracies of the four Maxar satellites used in our study are expressed in circular error probabilities or CE90 units, meaning that a given point will be within a specific radius 90% of the time, and in terms of RMSEs, which are one standard deviation of the residuals or distance-prediction errors. Our satellite data were resampled to a 50-cm spatial resolution for the panchromatic band and that band was used to panchromatically sharpen the NDVI to 50 cm. See also https://resources.maxar.com/data-sheets/quickbird; https://resources.maxar.com/data-sheets/geoeye-1; https://resources.maxar.com/data-sheets/worldview-2; and https://resources.maxar.com/data-sheets/worldview-3.