## [Peer Review File · Nature]

Manuscript Title: Sub-continental scale carbon stocks of individual trees in African drylands

Reviewer Comments & Author Rebuttals

Reviewer Reports on the Initial Version:

Referees' comments:

Referee #1 (Remarks to the Author):

Reviewer comments 398283_2_art_file_3850948_r84371

Towards continental scale monitoring of carbon stocks of individual trees in African drylands
Tucker et al.

The study constitutes an attempt to establish a framework for mapping carbon stocks at the level of individual trees in a large part of the African continent where accurate data for countries to respond to their international commitments is lacking. This is a real contribution in that direction. The good thing is that the tool can be improved with time for more accuracy even though permanent plots for continuous monitoring remains a challenge. Therefore, this paper serves two causes: scientific tool development and awareness raising for the need of continuous monitoring. There are several issues to be fixed (see comments below) and the paper will be ready for publication with the hope that the momentum to keep improving the framework will be sustained.

Introduction:

Lines 51-53: Is it your own definition of tree? If so, say it or provide reference (s)

Lines 59-62: Any citation for this point? cite the studies which have been conducted in drylands.

From lines 59-90 As there is not citation, this looks like anecdotal testimonies or report. The background in the introduction long yet the literature review is weak and unrepresentative. Please pay more attention to remote sensing (RS) application on forest/savanna AGB mapping. Please summarize previous studies about forest/savanna AGB mapping based on RS, like sensors and methods. Then gaps of previous studies to be resolved by this study should be put forward.

Vegetation carbon mapping based on multi-temporal images of Sentinel 1 and 2 have been published for West African dryland forest. But it remains some challenges such as mapping of individual tree crowns. That needs to be clearly presented with an appropriate flow.

Lines 101-103 Do you mean identify the species name? Not sure the reference quoted has reached that level.

Lines 122-113 How these 30 species were selected and why these? Or are you referring to individuals?

Lines 113-116 Unclear the rationale of the comparisons with equations established for wetter climatic zones

Results:

Line 129: Excluding individuals of crown $< 3m^2$ contradicts your claim of providing more accurate estimates

Lines 127-136: Please report field carbon values, like range, CV, SD, mean, median in a table.

Lines 166-167: Hope uncertainties due to this approach will be discussed

Lines 178-179: Surprising as you have excluded individuals smaller than 3 m² in crown size

Discussion

Line 290: Your lowest value of the range 0.1 m² is smaller than 3 m² while in your methodology you have excluded individuals below that 3 m². How possible is this?

Line 295: Minor error in the measurement culminating to large divergence in carbon estimate as indicated on line 291. Then the consequence is not minor

Lines 298-299 Why the same equation if 30 species have been measured. What was the use of the measurements on these 30 species?

Lines 305-306 Livestock and bush fires

Line 315 Yes, see previous comment

Methods:

The description was mass like a lab report. It should be revised in a more concise way.

Line 489: A flow chat about the procedure will be helpful.

Line 516-532: A table about images information should be listed in this section due to multisource images used.

Lines 556-557: Please, specify the exact period when the data was collected. Early dry season would not help your readers.

Lines 560-562: This is not clear to me. What data did you collect? The position of the individual trees or the crown data?

Line 567: The error seems to be quite high, and I am a bit afraid about the results of the carbon mapping. Does this influence the accuracy of the model? Probably, the exact position of the tree will be affected, which will affect the carbon mapping, because the environmental variables will not be considered at the right place.

Line 586: Please, specify the software used for tree segmentation. I think most of the readers are not familiar with this.

Lines 580-591: December – March are not early dry season in Africa. This period corresponds to the proper dry season. Please, correct.

Lines 593-594: Among so many methods, the reasons why authors used convolutional neural network models should be explained in the introduction and methods.

Lines 594-596: How did you manage to distinguish the crowns of the trees that are fused/clumped? This is common in West Africa savanna and forest. Many segmentation software programs fail to separate these crowns. Explain to the readers how you managed to do this.

Line 616: When abbreviations were used, they should be full spelled before, like CHIRPS.

Line 618: 0-15 m² or 3-15 m²? See line 634

Lines 649-652: Different tree sample sizes for different tree compartments. What is the implication for the accuracy of your estimates for these compartments (wood, leaves, roots)? The method used to convert tree data into wood, root and leaf mass is not clear. What allometric equations have you developed? How did you assess the accuracy of these equations before using them? And why didn't the authors consider the mass of branches? All this information is crucial to assess the accuracy of the authors' work. Several equations have been developed in West Africa that the authors could use, but they have not. This raises many questions about the reliability of the authors' work.

Figure 2: Is the third image about Santoro or Globbiomass?

Line 777 Is it Blovet or Bouvet?

Figure 4 What represent the dots beyond 1000 mm in the rainfall legend?

Bayala Jules

Referee #2 (Remarks to the Author):

This manuscript addresses a large knowledge gap in our understanding of the variation in density and carbon storage of dryland trees. The authors have amassed an impressive dataset of nearly 10 billion trees spanning African drylands from high resolution satellite data, field data, and machine learning. The reported tree-by-tree values and publically available database should be of great value and use to a range of scientific disciplines and field practitioners. These data are crucial as a baseline to evaluate how dryland trees will respond to changing climate and land-use. The results are original and rigorous and the manuscript well-written. I have a number of comments/questions:

The "viewer" will be an outstanding resource, as noted for stakeholders, policy, farmers, etc. Have you considered making this interactive, e.g., where corrections or additional information could be submitted for individual trees by registered users. For example, it would be great if a farmer could submit a form adding the tree species or provide other information such as tree diameter and height. This could be a powerful tool for NGOs working with farmers for C accounting and payment for ecosystem services.

At no fault of the authors, the allometric equations used to estimate tree-by-tree carbon storage are based on a small sample of 30 species (equations under review). A similar challenge is faced for more humid tropical forests when using, e.g., the Chave et al. 2014 equations. How representative are these 30 species of species across the study area? How variable is crown area? Is variation in tree height important for C estimates, e.g., Feldpausch et al. 2012? Variation in crown depth? It would be interesting to see some of the results on variation and drivers in crown area as these could be important for future studies, e.g., estimating LAI, transpiration or provision of shade for grazing animals or microclimate variation at sites.

The breakdown of C by humidity zone is interesting but more could be done. Could you also add additional analyses of drivers of the variation in C, e.g., soil type, precipitation seasonality (e.g., MCWD or another drought index), etc.? Why focus on annual precipitation? Would precip seasonality be more important? The comparison to DGVMs and other estimates is useful but it would be interesting to understand some of the drivers beyond precipitation.

Please report the uncertainty associated with each value, e.g., confidence intervals for Mg C ha⁻¹.

The root and leaf mass is a simple fraction of the above ground mass based on the allometric equations? If yes, how does root mass percentage vary? I think you need to be clearer about how the different C fractions are estimated if you are going to report the fractions.

Did you report the time period the measurements cover in the main text? The conclusion suggests you don't have time series data, although at line 248 you indicate you have time series data at least for some individual sites from 2002 to 2021. The methods report you used images from 2002 to 2020. How does this range of dates affect your estimates (e.g., tree mortality, recruitment, and growth during that period)? Over what period does the final mosaic map cover (if the map covers a range of years, how did you account for variation in the years across the mosaic)? I realize including a full-scale time series analysis is beyond the current focus of the manuscript and I look forward to seeing that work, e.g., could then use the to assess changes in tree density and carbon storage (e.g., for comparison to pantropical estimates such as Hubau et al. 2020 and looking at drivers of long-term patterns in tree loss and growth)

Figure 1, in (a) light colors are higher values but in (b) they are lower and this is somewhat confusing as I expected lighter colors in (b) have higher C storage.

Why is there a decline in Mg C ha⁻¹ at the highest precip values? I might expect a plateau at higher precipitation where you begin to see multiple canopy layers and canopy overlap. But why a decline?

Figure 4 appears to show tree crowns that were not detected. What were your commission and omission errors? I see you have included an uncertainty analysis in the SI; however, some of this information should also be included in the main text (e.g., also including confidence intervals in estimates).

Referee #3 (Remarks to the Author):

A. The authors present a method for mapping individual trees in African drylands using a Deep Learning framework that was applied to very high resolution optical satellites. The result is a map of individual tree crowns with associated biomass values of different tree compartments, that were predicted using allometric models with crown size as independent variable. Total biomass values were calculated by simply adding up all identified trees and compared to previous studies and ecosystem models.

B. On this scale (continental) the study is original and significant as it explicitly utilizes the characteristics of trees growing under open conditions (trees outside forests), where other studies often fail by relying on assumptions/models from forestry. The presented approach has great potential for monitoring tree growth in the area and studying the impact of climate change on the natural resources (local to continental).

C. The approach is valid using appropriate data of good quality. The presentation is very accessible.

D. The statistics are sound in general and uncertainties were assessed adequately.

E. Given the natural conditions of tree growth in the dry lands (early dry season, open and scattered), the method seems to be robust and reliable. The authors indicate adding a temporal component to their wall-to-wall map in further studies, which hopefully will proof this.

F. A few minor suggestions are provided at the end using line, figure and table numbers to indicate the position in the text.

G. The essential references were cited and extensive credit to previous work was given.

H. No further complain here: All sections of the manuscript are very clear and nothing is out of context.

Additional comments:

Line 99: Remove „...reducing water availability...“, seems to be superfluous

Line 110: I would prefer if you simply replace “estimating” with “for”. In a strict sense “estimating” can only be used when you apply a statistical estimation framework. The parameters of allometric model are estimated by some model fitting technique or population parameters are estimated using the Horvitz-Thompson estimator, etc. Models predict values, not estimate.

Line 115-116: Uncertainty of 19.75 % - What does that mean? Is it a confidence interval or more like a misclassification error calculated against some ground reference data? A small example to help interpretation of this number would be helpful, even though it is explained in the supplementary document.

Fig. 1 a): The upper right legend has two scales: Mg C / ha and, I assume, Mg Dm / ha. I would remove the dry mass scale as it provides no additional information. To convert from DM to C a fixed factor is used (0.47). Further, from the labelling in the legend, the unit of DM is unclear.

Line 161: remove “a” in the parentheses

Lines 170-171: “...except for reference, ...” I don’t get it. Do you mean that the cited study is the exception or is the comma wrongly placed?

Lines 188-189: “...rainfall-tree density/cover relationship...” This seems to be a mistake.

Line 192: “88%” – There should be a space between number and unit.

Line 283: “...except for ref47” – Similar to the previous comment at Lines 170-171. Seems to be the way of direct referencing in Nature publications? Hopefully, it gets resolved during publication.

Lines 291-292: From what I understand you developed own allometric equations based on field data from dryer areas with smaller trees. I'm not familiar with tree growth in this area but if your models are developed for trees that typically grow under open conditions, then there might be a bias for trees growing in more stand-like surroundings with tree-to-tree competition. Under the latter conditions, trees tend to allocate more biomass to the stem than to the crown/foilage and would have smaller crowns in relation to their wood biomass. Your models would then result in too low biomass values for the larger trees in the more humid area (if they grow in stands, which I don't really know).

Line 521: Add reference to Extended Data Fig. 8 to help assessing the timeliness of the data.

Lines 560-562: Please reformulate the sentence. Further, is it correct that only one person delineated 89,899 trees? Manually?

Line 858: Replace "than" with "that"

Table 1: Lower case/upper case not consistent

Line 881: Please check: "...with canopy of 3 m² or greater..." – superscript in the unit

Author Rebuttals to Initial Comments:

Referees' comments:

Referee #1 (Remarks to the Author):

Reviewer comments 398283_2_art_file_3850948_r84371

Towards continental scale monitoring of carbon stocks of individual trees in African drylands
Tucker et al.

The study constitutes an attempt to establish a framework for mapping carbon stocks at the level of individual trees in a large part of the African continent where accurate data for countries to respond to their international commitments is lacking. This is a real contribution in that direction. The good thing is that the tool can be improved with time for more accuracy even though permanent plots for continuous monitoring remains a challenge. Therefore, this paper serves two causes: scientific tool development and awareness raising for the need of continuous monitoring. There are several issues to be fixed (see comments below) and the paper will be ready for publication with the hope that the momentum to keep improving the framework will be sustained.

Authors: Thank you for your positive and constructive feedback.

Reviewer 1: Introduction:

Lines 51-53: Is it your own definition of tree? If so, say it or provide reference (s)

Authors: In the paper the word “tree” is used as a generic word for all “woody plants”: trees, shrubs, bushes, palm trees. The crown area threshold of 3 m² is just due to technical limitations due to the images resolution. Below this threshold some woody plants are mapped however an increasing proportion are missed as the crown area decrease that’s why we preferred to only consider woody plants above 3 m² of crown area. We have revised the sentence slightly to make it clear that the definition applies to this study.

Reviewer 1: Lines 59-62: Any citation for this point? cite the studies which have been conducted in drylands.

Authors: Thank you for your comment. We have now added several references to this part.

Reviewer 1: From lines 59-90 As there is not citation, this looks like anecdotal testimonies or report. The background in the introduction long yet the literature review is weak and unrepresentative. Please pay more attention to remote sensing (RS) application on forest/savanna AGB mapping. Please summarize previous studies about forest/savanna AGB mapping based on RS, like sensors and methods. Then gaps of previous studies to be resolved by this study should be put forward. Vegetation carbon mapping based on multi-temporal images of Sentinel 1 and 2 have been published for West

African dryland forest. But it remains some challenges such as mapping of individual tree crowns. That needs to be clearly presented with an appropriate flow.

Authors: Thanks for spotting this, indeed we missed to cite relevant studies in these paragraphs. We found the topic so attractive that we have done an entire review paper on it that summarized the current state of the art and the knowledge gaps, it is currently in press in the Journal of Remote Sensing. Consequently, we would prefer keeping references concerning the remote sensing technical background out of this Nature paper and refer to our other review work. We have however added several references related to the non-technical and applied aspects of woody cover mapping, as requested by the reviewer.

Lines 101-103 Do you mean identify the species name? Not sure the reference quoted has reached that level.

Authors: The species of the woody plant is not identified by these very high resolution images, the manuscript text just say "...identify isolated trees and map their crown area". It was not done either in the paper in reference (Brandt et al. 2020). Only a few species which particular crown shape or phenology such as *Faidherbia albida* (Lu et al 2022), or perhaps the baobab tree *Adansonia digitata* and probably the palm trees could be identified as species.

(Lu T., Brandt M., Tong X., Hiernaux P., Leroux L., Ndao B., Fensholt R. 2022. Mapping the abundance of multipurpose agroforestry *Faidherbia albida* trees in Senegal. *Remote Sens.* 14, 662. <https://doi.org/10.3390/rs14030662>).

Reviewer 1: Lines 122-113 How these 30 species were selected and why these? Or are you referring to individuals?

Authors: These 30 species were selected first in a sub-humid region of Mali (23 species, 647 woody plants) to encompass the most common species in order to assess the fodder mass available to pastoral livestock (in the framework of an ILRI research project on livestock production systems in central Mali (Wilson et al 1988). The objective was to cover at least 80% of the woody plant population in the Mopti administrative region of Mali (79,017km²). Then the data base was completed with woody plants selected in the arid region of Gourma in northern Mali (7 additional species, + 16 species already sampled, 514 woody plants) again to ensure at least 80% of the foliage mass of this arid to semi-arid region (150 to 500mm annual rainfall) (Hiernaux et al. 2009). The data base was also extended for wood and root masses especially in Mali and in a sub-humid area of Western Niger (Hiernaux et al. 2022). All together 900 woody plants of 30 species have been selected to build the foliage allometry, 698 of 30 species for the wood (trunk + branches + twigs) allometry, and only 26 of

6 species for the root (stump + coarse roots + fine roots) allometry. In all three, different growth forms (tall trees, short trees, shrubs and bushes are included).

We also restrict the study area to the region at the south of the Sahara desert recognized as sharing the same flora (White 1983).

Wilson T., Hiernaux P., McIntire J., 1988. Systems research in the West African arid and semi-arid zones: a synthesis of results from 1976 to 1986. ILCA, Addis Ababa, Ethiopia, 286p.

Hiernaux P., Diarra L., Trichon V., Mougine, E., Baup F., 2009, Woody plant population dynamics in response to climate changes from 1984 to 2006 in Sahel (Gourma, Mali). *Journal of Hydrology*, 375 (1-2): 103-113

Hiernaux P., Adamou Kalilou A., Kergoat L., Brandt M., Mougine E., Fitts Y., 2022. Woody plant decline in the Sahel of western Niger (1996-2017) : is it driven by climate or land use changes? *J. Arid Envir.*, 200: 104719

White, F. (1983) *The Vegetation of Africa, a Descriptive Memoir to Accompany the UNESCO/AETFAT/UNSO Vegetation Map of Africa (3 Plates, Northwestern Africa, Northeastern Africa, and Southern Africa, 1:5,000,000.* UNESCO, Paris.

Reviewer 1: Lines 113-116 Unclear the rationale of the comparisons with equations established for wetter climatic zones

Authors: This comparison is fully developed in the paper dedicated to allometry (ref 6 Hiernaux et al). The objective was to compare our allometry equations established with sole crown area as independent variable to 16 published allometry equations for the tropical angiosperms. Only two recently published reference (Djomo and Chimi 2017, Jucker et al 2017) include equations also based on crown area but for tropical forest in Cameroon and from a wide range of climates, all the other classically consider trunk diameter at breast height (DBH), tree height and wood density as independent variables. So the comparison required converting these equations to crown area allometry. We found our equations to fit within the range of equations and thus were more confident in their use in more humid area (beyond 600mm until 1000mm rainfall) in the study area. Please see the paper draft on allometry which we attach to the submission, as it is not yet published. Here all details are mentioned.

Results:

Line 129: Excluding individuals of crown < 3m² contradicts your claim of providing more accurate estimates

Authors: Even though we consider this work a major step forward, there are limits to what can be done from commercial remote sensing data captured at a distance of >600 km above the Earth surface. However, the contribution of these small trees to the area's carbon stocks and canopy cover is very small. We have tested this for areas where we are confident on the detection of small trees <3 m² but it did not change the overall numbers.

Reviewer 1: Lines 127-136: Please report field carbon values, like range, CV, SD, mean, median in a table.

Authors: In the referred paper (6 Hiernaux et al.) the allometry equations are described. It includes the carbon concentration (or gravity) in wood, leaves and roots set at 47% of dry mass. For the choice of this figure we refer to the chapter dedicated to carbon content woody plants in the IPCC 2006 report, selecting the figures given for tropical vegetation, especially the study by Feldpausch et al. 2004 which is referred to:

Feldpausch, T.R., Rondon, M.A., Fernandes, E.C.M., Riha, S.J., 2004. Carbon and nutrient accumulation in secondary forests regenerating on pastures in central Amazonia. *Ecological Applications* **14**: S164-S176.

Please see the following references which all confirm the choice made:

Yeboah, D., Burton, A.J., Storer, A.J. *et al.* Variation in wood density and carbon content of tropical plantation tree species from Ghana. *New Forests* **45**, 35–52 (2014). <https://doi.org/10.1007/s11056-013-9390-8>

Martin AR, Thomas SC. A reassessment of carbon content in tropical trees. *PLoS One*. 2011;6(8):e23533. doi: 10.1371/journal.pone.0023533. Epub 2011 Aug 17. PMID: 21858157; PMCID: PMC3157388.

Reviewer 1: Lines 166-167: Hope uncertainties due to this approach will be discussed

Authors: We have removed some parts of this analyses, and uncertainties are discussed in the referenced papers.

Reviewer 1: Lines 178-179: Surprising as you have excluded individuals smaller than 3 m² in crown size

Authors: Please see explanation above (Lines 51-53).

Discussion

Reviewer 1: Line 290: Your lowest value of the range 0.1 m² is smaller than 3 m² while in your methodology you have excluded individuals below that 3 m². How possible is this?

Authors: The numbers given here are from the field measurements.

Reviewer 1: Line 295: Minor error in the measurement culminating to large divergence in carbon estimate as indicated on line 291. Then the consequence is not minor

Authors: We have removed this part as it was not clear.

Reviewer 1: Lines 298-299 Why the same equation if 30 species have been measured. What was the use of the measurements on these 30 species?

Authors: As indicated above the allometry equations and their constructions are detailed in the reference (6 Hiernaux et al.). As species couldn't be identified in the very high resolution images, novel allometry equations had to be developed only relying on the crown area as independent variable regardless of the species. Data bases with 900 individuals for foliage mass and 698 for wood mass included 30 species among the most common tree and shrub species in ecosystems of the arid, semi-arid and sub-humid Sub-Saharan Africa were gathered to conduct this work. The available data on root mass limited the data base to 26 individual and 6 species.

Reviewer 1: Lines 305-306 Livestock and bush fires

Authors: Yes, livestock grazing has an impact on herbaceous mass, both due to the fodder intake and also because of trampling that in the wet season favor grass tilling and in the dry season turn standing straws into litter and accelerate the decomposition of the litter. Fire only occur in the dry season when the bulk of straws is large enough. Further south in perennial based savannas fire are used to facilitate the access of livestock to vegetative regrowth.

Reviewer 1: Line 315 Yes, see previous comment

Agreed, it's mentioned here, so no action was taken on the comment above.

Methods:

Reviewer 1: The description was mass like a lab report. It should be revised in a more concise way.

Authors: We have entirely revised the method part.

Reviewer 1: Line 489: A flow chart about the procedure will be helpful.

Authors: We agree, unfortunately we are limited by the number of display items and even had to remove a couple of supplementary figures.

Reviewer 1: Line 516-532: A table about images information should be listed in this section due to multisource images used.

Authors: We agree, please see Extended Data Table 2.

Reviewer 1: Lines 556-557: Please, specify the exact period when the data was collected. Early dry season would not help your readers.

Authors: The images were not taken at the same time, so we include now Extended Data Figure 7 and 9 showing the exact months the images were taken.

Reviewer 1: Lines 560-562: This is not clear to me. What data did you collect? The position of the individual trees or the crown data?

Authors: The formulation was misleading. In fact no data were collected but tree crowns were labelled on the screen. We have changed the formulation.

Reviewer 1: Line 567: The error seems to be quite high, and I am a bit afraid about the results of the carbon mapping. Does this influence the accuracy of the model? Probably, the exact position of the tree will be affected, which will affect the carbon mapping, because the environmental variables will not be considered at the right place.

Authors: Indeed these errors are high and the positions of the located trees are affected. But this should not impact on the numbers we provide. The uncertainty of the scene accuracy is not an issue when being compared to other environmental variables, as these have a spatial resolution being several orders of magnitude larger.

Reviewer 1: Line 586: Please, specify the software used for tree segmentation. I think most of the readers are not familiar with this.

Authors: We have used custom codes, which are made publicly available together with the paper.

Reviewer 1: Lines 580-591: December – March are not early dry season in Africa. This period corresponds to the proper dry season. Please, correct.

Authors: You are right in the Sahel the dry season starts in October, further south in the Sudanian zone it starts in November. However, we prioritized the early dry season when selecting the images whenever possible, see Extended Data Fig 7 and 9.

Reviewer 1: Lines 593-594: Among so many methods, the reasons why authors used convolutional neural network models should be explained in the introduction and methods.

Authors: Good point, but since we already published a pure method paper on this topic (Brandt et al 2020), we would like to keep this part as short as possible here. Nevertheless we have revised the entire method section.

Reviewer 1: Lines 594-596: How did you manage to distinguish the crowns of the trees that are fused/clumped? This is common in West Africa savanna and forest. Many segmentation software programs fail to separate these crowns. Explain to the readers how you managed to do this.

Authors: We did indeed solve this major issue, please see Brandt et al 2020 (Extended Data Fig 3) for more details. We have filled the gaps between tree crowns and gave these areas a higher weight during the model training. We have added a sentence on this in the revised method section.

Reviewer 1: Line 616: When abbreviations were used, they should be full spelled before, like CHIRPS.

Authors: We have corrected this.

Reviewer 1: Line 618: 0-15 m² or 3-15 m²? See line 634

Authors: You are right it should be corrected to 3-15m².

Reviewer 1: Lines 649-652: Different tree sample sizes for different tree compartments. What is the implication for the accuracy of your estimates for these compartments (wood, leaves, roots)? The method used to convert tree data into wood, root and leaf mass is not clear. What allometric equations have you developed? How did you assess the accuracy of these equations before using them? And why didn't the authors consider the mass of branches? All this information is crucial to assess the accuracy of the authors' work. Several equations have been developed in West Africa that the authors could use, but they have not. This raises many questions about the reliability of the authors' work.

Authors: Answer are already given in responses to questions lines 112-113, 113-116, 298-299 and are fully developed in the paper ref 6.

The allometry equations have been developed independently for foliage and wood masses. One of the reasons is that foliage had to be harvested at the seasonal peak mass, which was not always convenient for wood destructive work. Same applies to root excavation with foliage. This independence actually facilitate the calculation of approximation error per tree (root square of the sum of square errors) calculated to assess the prediction accuracy of the allometry equations.

Yes branches, coarse, fine and twigs were all included together with trunk or stems into the wood mass.

It is true that several allometry equations have been developed for woody plants in West Africa, but most of them are developed per species and consider the diameter at breast height (DBH), often associated to tree height and wood density, as independent variables. These variable are not accessible from satellite images. Only in two recent papers we found equations based on crown area, one for trees in humid tropical forest of Cameroon and the other with data coming from all type of climate (but not from arid-semi arid tropics). Therefore we had to develop novel equations that we compared to 16 published ones for the tropics, in general more humid.

Reviewer 1: Figure 2: Is the third image about Santoro or Globbiomass?

Authors: To our knowledge, these are the same: https://globbiomass.org/wp-content/uploads/GB_Maps/Globbiomass_global_dataset.html

Reviewer 1: Line 777 Is it Blovet or Bouvet?

Authors: Bouvet, thanks for spotting this.

Reviewer 1: Figure 4 What represent the dots beyond 1000 mm in the rainfall legend?

Authors: The legend has moved, thanks for spotting this.

Bayala Jules

Referee #2 (Remarks to the Author):

This manuscript addresses a large knowledge gap in our understanding of the variation in density and carbon storage of dryland trees. The authors have amassed an impressive dataset of nearly 10 billion trees spanning African drylands from high resolution satellite data, field data, and machine learning. The reported tree-by-tree values and publically available database should be of great value and use to a range of scientific disciplines and field practitioners. These data are crucial as a baseline to evaluate how dryland trees will respond to changing climate and land-use. The results are original and rigorous and the manuscript well-written. I have a number of comments/questions:

The "viewer" will be an outstanding resource, as noted for stakeholders, policy, farmers, etc. Have you considered making this interactive, e.g., where corrections or additional information could be submitted for individual trees by registered users. For example, it would be great if a farmer could submit a form adding the tree species or provide other information such as tree diameter and height. This could be a powerful tool for NGOs working with farmers for C accounting and payment for ecosystem services.

At no fault of the authors, the allometric equations used to estimate tree-by-tree carbon storage are based on a small sample of 30 species (equations under review). A similar challenge is faced for more humid tropical forests when using, e.g., the Chave et al. 2014 equations. How representative are these 30 species of species across the study area? How variable is crown area? Is variation in tree height important for C estimates, e.g., Feldpausch et al. 2012? Variation in crown depth? It would be interesting to see some of the results on variation and drivers in crown area as these could be important for future studies, e.g., estimating LAI, transpiration or provision of shade for grazing animals or microclimate variation at sites.

Authors: We will explore ways of simplifying access to our output data of the 10 B trees with the viewer and understand the importance of doing this.

Authors: The 30 species in the allometry were selected first in a sub-humid region of Mali (23 species, 647 woody plants) to encompass the most common species in order to assess the fodder mass available to pastoral livestock (in the framework of an ILRI research project on livestock production systems in central Mali (Wilson et al 1988). The objective was to cover at least 80% of the woody plant population in the Mopti administrative region of Mali (79,017 km²). Then the data base was completed with woody plants selected in the arid region of Gourma in northern Mali (7 additional species, + 16 species already sampled, for a total of 514 woody plants) again to ensure at least 80% of the foliage mass of this arid to semi-arid region (for the 150 to 500 mm annual rainfall region) (Hiernaux et al. 2009). The data base was also extended for wood and root masses in Mali and in a sub-humid area of Western Niger (Hiernaux et al. 2022). All together, 900 woody plants of 30 species have been selected to build the foliage allometry, 698 of 30 species for the wood (trunk + branches +

twigs) allometry, and only 26 of 6 species for the root (stump + coarse roots + fine roots) allometry. In all three, different growth forms (tall trees, short trees, shrubs and bushes are included).

We also restrict the study area to the region at the south of the Sahara desert recognized as sharing the same flora (White 1983).

Is variation in height important for C estimates?

Authors: Yes, tree height in addition to crown area would improve the quality of the fit of allometry equation, and reduce the approximation error on prediction. We have developed them in the allometry paper (ref 6) but unfortunately for the time being, we cannot retrieve tree height with sufficient precision from the very-high resolution images. Thus we are constrained to use the sole crown area as independent variable. When better tree height data eventually becomes available we will use it to improve our carbon numbers.

We had a thorough evaluation using the shadows cast by trees and the length of the shadow from the tree crown centroid to the maximum length of the shadow for several hundred trees, using the satellite's specific time of imaging & viewing perspective. However, the uncertainty of this method of tree height determination was substantial and we concluded it was not a viable method of tree height approximation.

We agree fully on your points about the interest of the variations and their drivers to widen the scope of application to estimate LAI, transpiration...

Reviewer 2: The breakdown of C by humidity zone is interesting but more could be done. Could you also add additional analyses of drivers of the variation in C, e.g., soil type, precipitation seasonality (e.g., MCWD or another drought index), etc.? Why focus on annual precipitation? Would precip seasonality be more important? The comparison to DGVMs and other estimates is useful but it would be interesting to understand some of the drivers beyond precipitation.

Authors: We agree with reviewer #2 observation that more could be done with other drivers of carbon variation. We are working on disaggregating our results by sandy and non-sandy soils. We are also attempting to work in more local areas to determine the effects of climate and land use change upon tree cover. See Hiernaux et al. 2022 above. However, we are limited by how much additional material can be packed into our article and have already started additional studies to elaborate and explore some of our findings. By making our output data available to all, we hope to stimulate open-access uses of our data.

Reviewer 2: Please report the uncertainty associated with each value, e.g., confidence intervals for Mg C ha⁻¹.

Authors: We evaluated the uncertainty of our tree crown area mapping and carbon estimation in several ways, which is described in the revised Methods section. First, we quantify our tree crown mapping omission and commission errors by inspecting randomly selected areas from UTM Zones 28 to 37, validating that our neural network generalizes over UTM zones consistently. Second, we quantify the relative error of our carbon estimation due to the allometric equations as well as the relative error in carbon estimation due to errors by the neural network is computed on external validation data from. For the second analysis, we followed Bevington and Robinson 2003. This error analysis focuses on the absolute differences in the measured and the predicted quantities. We used the Bevington and Robinson approach because we wanted to avoid wrong assumptions on the distribution of the data and the errors (assuming Gaussian distributions, the report uncertainties could be turned into confidence intervals – we strongly prefer not to make this assumption). Third, we also analyses the area prediction error at the level of individual trees of certain sizes in terms of the root-mean-square error, see the new Extended Data Figure 8 (which is proportional to confidence intervals when assuming normality).

Reviewer 2: The root and leaf mass is a simple fraction of the above ground mass based on the allometric equations? If yes, how does root mass percentage vary? I think you need to be clearer about how the different C fractions are estimated if you are going to report the fractions.

Authors: Actually the root mass and foliage mass are assessed independently of the wood mass. Yet the individual allometry equations results in ratios between Root and Wood mass that typically start quite high in small trees or shrubs (close to 1) and decrease fast as crown area increase to reach an asymptote between 0.2 and 0.3 in large trees. The foliage to wood ratio follows the same type of trend but at much lower value, the foliage reaching only a few 3 to 5% of the wood mass in large trees.

Reviewer 2: Did you report the time period the measurements cover in the main text? The conclusion suggests you don't have time series data, although at line 248 you indicate you have time series data at least for some individual sites from 2002 to 2021. The methods report you used images from 2002 to 2020. How does this range of dates affect your estimates (e.g., tree mortality, recruitment, and growth during that period)? Over what period does the final mosaic map cover (if the map covers a range of years, how did you account for variation in the years across the mosaic)? I realize including a full-scale time series analysis is beyond the current focus of the manuscript and I look forward to seeing that work, e.g., could then use the to assess changes in tree density and carbon storage (e.g., for comparison to pantropical estimates such as Hubau et al. 2020 and looking at drivers of long-term patterns in tree loss and growth)

Authors: Yes, the time period of our satellite data used for the analysis was 2002 to 2020. Because we were able to access the satellite data used in our paper at no-charge within the US Government, we were able to use 326,523 DigitalGlobe satellite images. We have since added an additional 115,000 satellite images to our study from 2020, 2021, and 2022, bringing the total number of images to

440,000. We fully agree with the reviewer that a comparison between different time periods would be extremely interesting and plan to pursue this once this mapping benchmark paper is completed.

And, yes we agree with reviewer that it was not clear that what was presented in line 248 (Figure 5) was in fact an example of how what we have developed can be used to monitor changes over time, even we don't do this for the wall-to-wall mapping. The text has been slightly revised to make this point clearer and should hopefully reduce the confusion about this point.

Figure 1, in (a) light colors are higher values but in (b) they are lower and this is somewhat confusing as I expected lighter colors in (b) have higher C storage.

Authors: It may look confusing but we have tested multiple sets of color schemes and decided that this one is the clearest one.

Why is there a decline in Mg C ha⁻¹ at the highest precip values? I might expect a plateau at higher precipitation where you begin to see multiple canopy layers and canopy overlap. But why a decline?

Authors: We have discussed this decline but a similar tendency has been reported by L-VOD studies, which gives a certain confidence. We have investigated our data for these areas and find no artifacts such as shading to explain it. It is possible that land cover change for agricultural development is more pronounced in this zone than elsewhere. This is a topic we will pursue subsequent to publication of our paper and encourage others to do the same.

Between 800 and 1000 mm annual rainfall isohyets it is still savannas and woodlands that prevail, forest are limited to narrow riverine forests, so it is not so much the overlap between crown but the land use (largely cleared croplands) and perhaps the soil fertility (large extends of indurated ferrallitic soils) that explain the plateau and small decline.

Reviewer 2: Figure 4 appears to show tree crowns that were not detected. What were your commission and omission errors? I see you have included an uncertainty analysis in the SI; however, some of this information should also be included in the main text (e.g., also including confidence internals in estimates).

Authors: Yes, not 100% of the trees can always be detected. We have done an extensive evaluation of omission and commission errors. Extended Data Fig. 5, which summarizes our omission and commission errors, shows approximately even omission & commission errors for the 3-15 m² tree class—260 and 240 values from our 50,570 tree evaluation, respectively. We have discussed of moving some of these information to the main text, but given the word limit, it remains in the method section and extended data part, which is part of the paper.

Addressing referee #2 questions

Authors: We include our allometric data for every tree foliage, tree wood, and tree roots with their associated crown areas at the ORNL Data Repository in Tennessee USA with a longer write up there.

We've also added this summary text of 226 words to our Methods section under the sub-heading of Allometry. One of the deciding variables on the consistency and applicability of the allometry are the root-to-wood ratios and foliage-to-wood ratios over the range of our crown areas, as well as estimates of prediction uncertainty.

Very high-resolution satellite images and deep learning have achieved mapping of individual trees over large areas¹. Each tree is georeferenced in the satellite data and defined by crown area. This enables the use of allometric equations for foliage, wood and root dry masses or carbon based on crown area regardless of species. This was met by reanalyzing existing Sahelian and Sudanian woody plant data from destructive sampling. Overall, the seasonal maximum foliage, wood, and root dry masses were measured on 900, 698, and 26 trees or shrubs from 27, 26, and 5 species respectively, where crown area was also measured. The allometric regression equations for foliage, wood, or root masses are power functions and are independent of species. All the regression outputs were inter-compared for fit indicators, by systematic estimates of prediction uncertainty, as well as root-to-wood ratios and foliage-to-wood ratios over the range of crown areas. This resulted in a set of ordinary least square log-log equations with crown area as independent variable. The Sahelian and Sudanian allometry equations were also compared to published allometry equations for tropical trees, primarily from more humid tropics, that are generally based on stem diameter, tree height, and wood density. Our allometric predictions are within the range of other allometry predictions, reinforcing the confidence in their use beyond the Sahelian and Sudanian domains into sub-humid savannas for discrete trees¹⁹.

We have a conundrum: the more we explain about our allometry the longer the paper gets. Our allometry paper has been conditionally accepted and the allometry team returned their revisions to Forest Ecology and Management 2 weeks ago. We can easily accommodate more details with the tabular allometry for our tree crown data at the Oak Ridge National Laboratory Data Center in Tennessee USA. We are finalizing our data write up in the next fortnight for our Allometry and Data at <https://doi.org/10.3334/ORNLDAAAC/2117>. Please advise what you suggest we should relocate there.

The question from reviewer #2 about our carbon conversion term of 0.47:

We have added the reference of "McGrouddy, M.E., Daufresne, T., & Hedin, L. Scaling of C:N:P stoichiometry in forests worldwide: implications of terrestrial Redfield-type ratios. Ecology 85(9), 2390-2401 (2004)", as reference #46 in the Methods Section, documenting the 0.47 conversion term.

Referee #3 (Remarks to the Author):

A. The authors present a method for mapping individual trees in African drylands using a Deep Learning framework that was applied to very high resolution optical satellites. The result is a map of individual tree crowns with associated biomass values of different tree compartments, that were predicted using allometric models with crown size as independent variable. Total biomass values were calculated by simply adding up all identified trees and compared to previous studies and ecosystem models.

B. On this scale (continental) the study is original and significant as it explicitly utilizes the characteristics of trees growing under open conditions (trees outside forests), where other studies often fail by relying on assumptions/models from forestry. The presented approach has great potential for monitoring tree growth in the area and studying the impact of climate change on the natural resources (local to continental).

C. The approach is valid using appropriate data of good quality. The presentation is very accessible.

D. The statistics are sound in general and uncertainties were assessed adequately.

E. Given the natural conditions of tree growth in the dry lands (early dry season, open and scattered), the method seems to be robust and reliable. The authors indicate adding a temporal component to their wall-to-wall map in further studies, which hopefully will proof this.

F. A few minor suggestions are provided at the end using line, figure and table numbers to indicate the position in the text.

G. The essential references were cited and extensive credit to previous work was given.

H. No further complain here: All sections of the manuscript are very clear and nothing is out of context.

Additional comments:

Line 99: Remove „...reducing water availability...“, seems to be superfluous

Authors: we agree and this has been deleted.

Reviewer 3: Line 110: I would prefer if you simply replace “estimating” with “for”. In a strict sense “estimating” can only be used when you apply a statistical estimation framework. The parameters of allometric model are estimated by some model fitting technique or population parameters are estimated using the Horvitz-Thompson estimator, etc. Models predict values, not estimate.

Authors: we agree and “estimating” has been changed to “predict”.

Reviewer 3: Line 115-116: Uncertainty of 19.75 % - What does that mean? Is it a confidence interval or more like a misclassification error calculated against some ground reference data? A small example to help interpretation of this number would be helpful, even though it is be explained in the supplementary document.

Authors: 19.8% is the relative uncertainty when we consider the error in mapping of trees crown areas combined with the error in conversion of trees crown areas to drymass and carbon. We now

have an uncertainty section in the Methods section. The errors were computed based on reference data, where different independent data sets were combined. We followed Bevington and Robinson 2003 (ref. 55). The errors are differences between observed and true measurements, and the uncertainties are the magnitudes of the errors [ref. 55, p.14]. Now the relative uncertainty expresses the uncertainty by a percentage. For example, if the relative uncertainty is 10% and the measurement is 200, then we can expect an error of magnitude 20.

Reviewer 3: Fig. 1 a): The upper right legend has two scales: Mg C / ha and, I assume, Mg Dm / ha. I would remove the dry mass scale as it provides no additional information. To convert from DM to C a fixed factor is used (0.47). Further, from the labelling in the legend, the unit of DM is unclear.

Authors: We have removed the DM per ha scale.

Reviewer 3: Line 161: remove “a” in the parentheses

Authors: done and thank you for pointing this out.

Reviewer 3: Lines 170-171: “...except for reference, ...” I don’t get it. Do you mean that the cited study is the exception or is the comma wrongly placed?

Authors: This has been corrected and thank you for pointing this out.

Reviewer 3: Lines 188-189: “...rainfall-tree density/cover relationship...” This seems to be a mistake.

Authors: Corrected to “rainfall-tree density relationship”.

Reviewer 3: Line 192: “88%” – There should be a space between number and unit.

Authors: done.

Reviewer 3: Line 283: “...except for ref47” – Similar to the previous comment at Lines 170-171. Seems to be the way of direct referencing in Nature publications? Hopefully, it gets resolved during publication.

Authors: This is not in our hands.

Reviewer 3: Lines 291-292: From what I understand you developed own allometric equations based on field data from dryer areas with smaller trees. I’m not familiar with tree growth in this area but if

your models are developed for trees that typically grow under open conditions, then there might be a bias for trees growing in more stand-like surroundings with tree-to-tree competition. Under the latter conditions, trees tend to allocate more biomass to the stem than to the crown/foilage and would have smaller crowns in relation to their wood biomass. Your models would then result in too low biomass values for the larger trees in the more humid area (if they grow in stands, which I don't really know).

Authors: Yes, you are right the allometric equations were developed based on trees and shrubs representative of the arid and semi-arid zones from 150 to 600mm annual rainfall where they generally grow isolated or in small clumps, some in dense thickets such as in the “tiger bush” (Hiernaux et Gérard 1999; Trichon et al. 2018). Between 800 and 1000 mm annual rainfall isohyets it is still savannas and woodlands that prevail, forest are limited to narrow riverine forests, so it is not so much the overlap between crown that makes a difference. Yet the size of the largest trees is larger as mean rainfall increase so that we were not sure that we could use our equations beyond the 600 mm rainfall isohyet. So we have compared the predictions of our equations to the predictions of 16 published allometry equations for the tropical angiosperms. Only two recently published reference (Djomo and Chimi 2017, Jucker et al 2017) include equations based on crown area as ours but the first for humid tropical forest in Cameroon and from the second for a wide range of climates but not arid and semi-arid tropics. All the other equations classically consider trunk diameter at breast height (DBH), tree height and wood density as independent variables not retrievable from satellite images. So the comparison required converting these equations to crown area allometry. We found the prediction of our equations to fit within the range of the predictions of other equations, and be very close to some of them such as Chave et al. 2005, Henry et al 2009, Mukuralinda 2021, Brown 1997, Ogawa 1965 and Kuyah et al 2012 all established in more humid tropics. Thus were more confident in the use of our equations in more humid area (beyond 600mm until 1000mm rainfall) of the study area.

Hiernaux P. & B. Gérard, 1999. The influence of vegetation pattern on the productivity, diversity and stability of vegetation: the case of the ‘brousse tigrée’ in the Sahel. *Acta Oecologica*, 20,3: 147-158.

Trichon V, Hiernaux P, Walcker R, Mougin E. , 2018. The persistent decline of patterned woody vegetation: The tiger bush in the context of the regional Sahel greening trend. *Glob Change Biol*. 00:1–16. <https://doi.org/10.1111/gcb.14059>

Reviewer 3: Line 521: Add reference to Extended Data Fig. 8 to help assessing the timeliness of the data.

Authors: Extended Data Fig. 8 summarizes the timing of all the imagery used in the analysis and we are not sure we understand what reviewer is asking for here.

Reviewer 3: Lines 560-562: Please reformulate the sentence. Further, is it correct that only one person delineated 89,899 trees? Manually?

Authors: Thanks. This sentence now reads “Our training data were collected by one team member and are a carefully selected manual delineation of 89,899 individual trees under a range of atmospheric conditions, viewing perspectives, and ecological settings.”

Reviewer 3: Line 858: Replace “than” with “that”

Authors: done

Reviewer 3: Table 1: Lower case/upper case not consistent

Authors: corrected

Reviewer 3: Line 881: Please check: “...with canopy of 3 m² or greater...” – superscript in the unit

Authors: Corrected.

Reviewer Reports on the First Revision:

Referees' comments:

Referee #1 (Remarks to the Author):

The comments we have raised on the first version have been adequately addressed and that has tremendously improved the clarity of the methods, precisions of the estimates and the messages. This is an original work and truly needed. I'm sure it will contribute a lot in facilitating the evaluation of tree planting operations in drylands and savanna. I'm entirely satisfied with the way all our commented s have been addressed. I therefore strongly recommend this manuscript for publication.

Referee #2 (Remarks to the Author):

The authors have addressed the questions raised in my original review in both their written response to my review and the revised manuscript and supplemental information. I have no additional questions for the authors. Their revisions do not change the main presentation and interpretations of the data and conclusions. Therefore, the manuscript continues to address a large knowledge gap in our understanding of the variation in density and C storage of dryland trees and the results are rigorous, original and well-presented.